# The *Hydractinia* cell atlas reveals cellular and molecular principles of cnidarian coloniality

David A. Salamanca-Díaz[1,2,3,14], Helen R. Horkan [4,9,14] ✉,
Helena García-Castro [1,2,3], Elena Emili[1,10], Miguel Salinas-Saavedra[4],
Alberto Pérez-Posada[1,2,3], Maria Eleonora Rossi[4,5], Marta Álvarez-Presas [5,11],
Rowan Mac Gabhann [4], Paula Hillenbrand [4], Febrimarsa[4,12],
Camille Curantz[4,13], Paris K. Weavers [4], Yasmine Lund-Ricard [4],
Tassilo Förg [6], Manuel H. Michaca[7], Steven M. Sanders [7], Nathan J. Kenny [8],
Jordi Paps [5], Uri Frank [4] ✉ & Jordi Solana [1,2,3] ✉

Coloniality is a widespread growth form in cnidarians, tunicates, and bryozoans, among others. Colonies function as single physiological units despite their modular structure of zooids and supporting tissues. A key question is how structurally and functionally distinct colony parts are generated. In the cnidarian *Hydractinia symbiolongicarpus*, colonies consist of zooids (polyps) interconnected by stolons attached to the substrate. Using single-cell transcriptomics, we profiled ~200,000 *Hydractinia* cells, including stolons and two polyp types, identifying major cell types and their distribution across colony parts. Distinct colony parts are primarily characterised by unique combinations of shared cell types and to a lesser extent by part-specific cell types. We identified cell type-specific transcription factors (TFs) and gene sets expressed within these cell types. This suggests that cell type combinations and occasional innovations drive the evolution of coloniality in cnidarians. We uncover a novel stolon-specific cell type linked to biomineralization and chitin synthesis, potentially crucial for habitat adaptation. Additionally, we describe a new cell type mediating self/non-self recognition. In summary, the *Hydractinia* cell atlas provides insights into the cellular and molecular mechanisms underpinning coloniality.

Coloniality[1] is a growth form that occurs in diverse animal groups including cnidarians[2], tunicates[3], and bryozoans[4], among others. Contrary to solitary animals, colonial organisms are modular, consisting of individual, clonal zooids that are connected by live tissue, sharing a vascular network, nervous system, and migratory cells[5–8]. Growth and reproduction are mostly integrated in colonial animals; therefore, a single colony functions as a meta-individual, being a single physiological unit. Many colonial animals, such as ascidians and cnidarians, exhibit robust regenerative capacities that are based on populations of adult stem/stem-like cells[5,9,10]. In *Hydractinia*, these adult stem cells are the pluripotent i-cells[11]. Zooids within a colony may be polymorphic, i.e.

they become specialised in one or more functions. To protect their genetic uniformity, many colonial animals possess a genetically based allorecognition system, allowing them to discriminate between self and non-self with a high degree of precision. It is thought that allorecognition evolved to prevent germ cell parasitism between conspecifics[12]. Allorecognition genes have been identified in cnidarians and tunicates, and shown to be phylogenetically unrelated to each other and to vertebrate MHC/T-cell-mediated allorecognition[13,14]. Finally, colonial growth has evolved and been lost multiple times, even within a single phylum[2]. Despite the many interesting features of animal colonies, the cellular and molecular mechanisms that govern them remain unknown.

Members of the genus *Hydractinia*[15] form colonies that grow on gastropod shells inhabited by hermit crabs. A single, sexually produced larva colonises the shell and metamorphoses to generate a primary zooid (called polyp) and gastrovascular tubes that are attached to the substratum and known as stolons. Stolons develop into a dense network that continuously buds new, genetically identical but polymorphic polyps. A colony typically possesses several polyp types, each specialised in specific function such as prey capture (feeding polyp or gastrozooid), production of gametes (sexual polyps or gonozooid), or defence (dactylozooids). This growth process relies on a lineage of adult, migratory pluripotent stem cells known as i-cells[5,11]. *Hydractinia symbiolongicarpus* (hereafter referred to as *Hydractinia*) is an established model organism that has been studied since the time of August Weismann in the 19th century[16] and throughout the 20th [17] and 21st centuries[15]. It is amenable to genetic manipulation, has a sequenced genome[18], multiple transcriptomic resources[19], and is easy to culture in the lab on glass slides that contain gastrozooids, gonozooids, and a stolonal network. Despite all cells being continuously derived from i-cells, it is unknown if different colony parts are made of the same cell types or if specific types are the building blocks of each specific colony component.

Recently, single-cell transcriptomics (sc-RNA-seq) has provided an opportunity to classify individual cells into distinct cell types by analysing their unique gene expression profiles and clustering them based on shared transcriptional patterns[20]. This approach offers unprecedented insights into cellular heterogeneity and complex biological systems[21,22]. Using single-cell transcriptomics, the cell-type atlases of numerous organisms, including several solitary and colonial cnidarians[23–31] have been profiled. Single-cell transcriptomics can reveal the cellular composition of the different parts of a colony by using part-specific libraries. However, previous single-cell studies on colonial cnidarians have been constructed from mixed-cell libraries and could not profile colony parts individually.

Here, we present a single-cell atlas of the individual colony parts of *Hydractinia*. We generated ~200 K cellular profiles from distinct samples, including feeding- and sexual polyps, and stolons. We characterised all major cell types, and the transcription factors (TFs) and gene networks that are expressed in them. One of our main findings is that most cell types are shared between colony parts, albeit in different proportions, and part-specific cell types were rare. This might explain the ease of gaining and losing coloniality over evolutionary time scales. We validated novel cell types by whole-mount imaging of cellular staining using both mRNA in situ hybridisation and immunofluorescence. One such validated cell type, *Alr1 +*, showed specific expression of the *Alr1* gene (a member of the allorecognition complex), which has previously been shown to be critical in self/non-self recognition[32]. Alr1 was localised to epidermal epithelial cells as expected, given that allogeneic contacts are established by this tissue. Moreover, we experimentally show that the epidermis, not the gastrodermis, is the tissue that can discriminate self from non-self, consistent with the expression of *Alr1* in its cells. Among the part-specific cell types, we uncovered one that is enriched in stolons and expresses biomineralization and chitin synthesis enzymes. We studied the transcriptome of this cell type, revealing a cluster of repeat-containing *Shematrin-like* and *Prisilkin-like* genes that resemble molluscan shell matrix proteins, raising the possibility that *Hydractinia* has co-opted a biomineralization gene programme to attach to the gastropod shells inhabited by hermit crabs, a key adaptation to their environment. Altogether, the *Hydractinia* cell atlas reveals the cellular and molecular principles of their colonial growth.

## Results

### A cell-type atlas of *Hydractinia*

We obtained cell suspensions of laboratory-grown *Hydractinia* colonies using ACME[33] with modifications (see "Methods"). We included mixed polyp samples, selected feeding polyps (gastrozooids) and reproductive polyps (gonozooids), as well as stolons (Fig. 1A). We then performed a total of 4 SPLiT-seq experiments, comprising 13 sublibraries, and sequenced these libraries using Illumina NovaSeq 6000 technology at 150 bp paired-end read length (Fig. 1B and Supplementary Data 1). We obtained a total of 6.9 billion reads.

We then used the *Hydractinia symbiolongicarpus* chromosome-level genome assembly[18] to map these reads using our SPLiT-seq read processing pipeline[33,34]. We further annotated the gene models using DIAMOND[35] (Supplementary Data 2). We used Muon[36] to build a multimodal matrix with uniquely mapped reads and with reads mapping to many loci, but used the matrix containing only singly mapping reads for the subsequent single-cell analysis. After further filtering steps, we obtained a matrix containing 199,113 cells. We used Harmony[37] to leverage batch effects from the different experiments, and built a kNN graph using 40 neighbours and 75 principal components.

As previously reported, SPLiT-seq delivers low numbers of reads and genes per cell[33,38–40] (Supplementary Fig. 1A, B). In spite of this, and thanks to the high number of cell profiles that can be obtained using SPLiT-seq, the Leiden algorithm for cell clustering at a resolution of 1.5 yielded a robust identification of 53 distinct cell clusters (Fig. 1C and Supplementary Fig. 1B, C). We generated marker genes for each cluster using both the Wilcoxon and Logistic Regression methods (Supplementary Data 3–5). We used PAGA to group these clusters in broad types (Fig. 1D). In addition, we performed a co-occurrence analysis on cell-type clusters[29]. This analysis corroborated our clustering results (Supplementary Fig. 1F), and helped with cluster annotation together with gene marker examination of previously published *Hydractinia* and cnidarian cell-type literature (Supplementary Note 1).

The *Hydractinia* cell-type repertoire included i-cells[11,24,41], the pluripotent stem cells that drive colonial growth and regeneration, committed progenitors, germ cells[42], as well as a panoply of differentiated cell types previously described in the literature, comprising neurones[24,30,43], nematoblasts and nematocytes[23,44], gland cells[45,46]. These have known localisation based on morphological description, histology, mRNA in situ hybridisation, immunofluorescence staining and transgenic reporter lines for specific marker genes (Fig. 1E and Supplementary Note 1). We additionally identified epithelial and epitheliomuscular cells and a number of previously unidentified cell types. Similar to other cnidarian cell atlases[25,27,29,30,47], the most abundant cell types were neurones, nematoblasts, nematocytes, epithelial and epitheliomuscular cells. We also identified a set of mucosal and digestive gland cells, as well as a novel cluster of *Conodipine+* cells, marked by the expression of venomous proteins, and including a cluster, the *Alr1+* cells, expressing allorecognition genes. We validated several of these cell types with in situ hybridisation using probes designed to target their marker genes (Supplementary Data 6) as well as immunofluorescence techniques. Gland cell cluster *Chitinase2+* cells (24) were validated by colorimetric in situ hybridisation and are located primarily in the gastrodermis of the feeding polyp and the stolon (Fig. 1F). Using SABER-FISH, we identify epithelial *Conodipine + /Avidin+* cells at the base of feeding polyp tentacles (Fig. 1G). Altogether, these results show that our single-cell data resolves the major cell types of *Hydractinia* polyps and stolons.

### Cellular composition of *Hydractinia* colony parts

We then aimed to ascertain the cellular composition of each individual colony part. We examined the contribution of each sample, including manually isolated stolons as well as sexual and feeding polyps, to our integrated analysis, revealing key differences in their cellular composition (Fig. 2A). In stolons, we observed markedly fewer gland cells, neurones, and germ cells; the latter were abundant in sexual polyps as expected. Furthermore, sexual polyps exhibited lower counts of

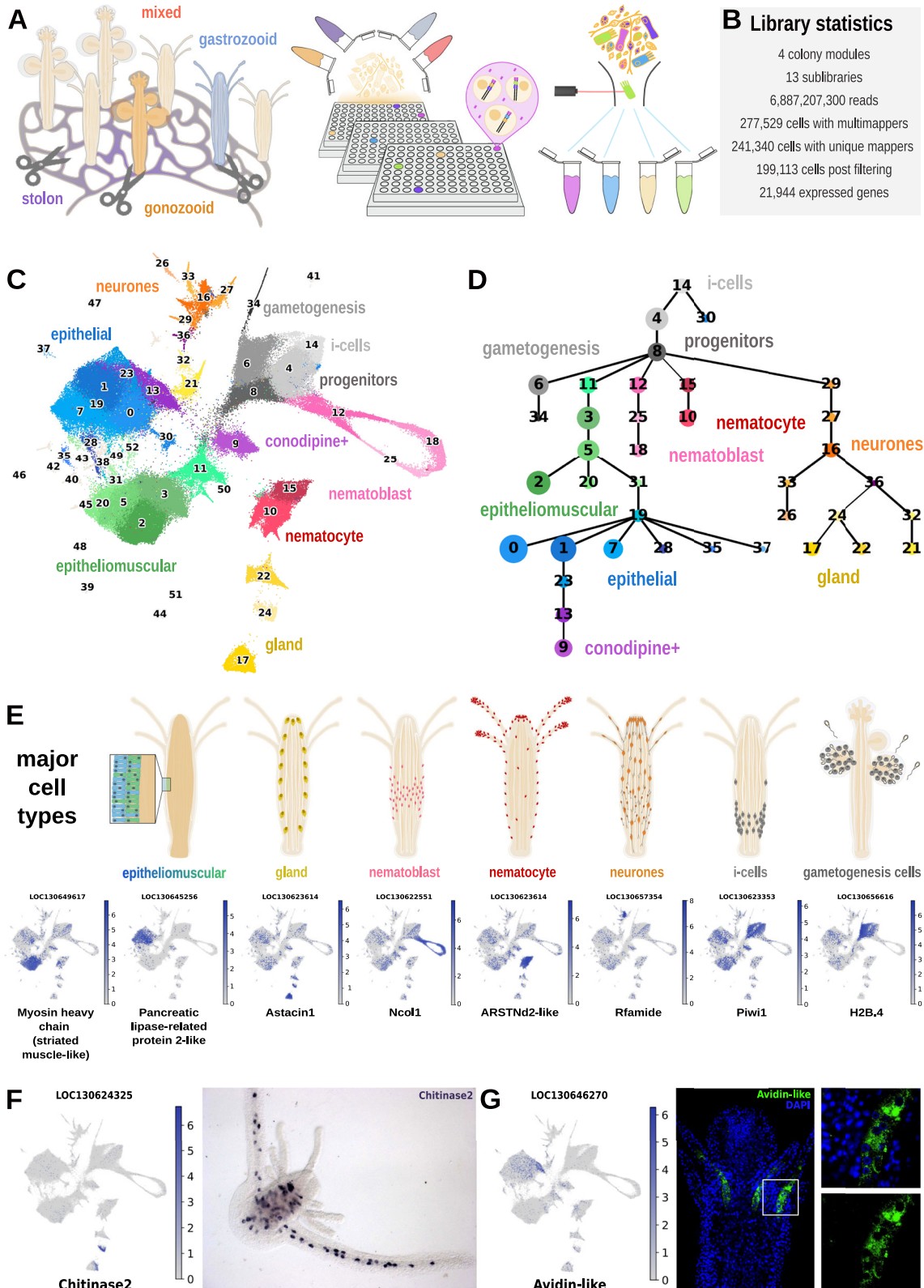

**Fig. 1 | Single-cell atlas of the colonial cnidarian *Hydractinia symbiolongicarpus*.**
**A** Scheme of experimental design to collect and process the samples. Detailing sample collection of sublibraries including mixed feeding and sexual polyps 'mixed', feeding polyps only 'feeding', sexual polyps only 'sexual' and stolon-only 'stolon'. **B** UMAP projection of the annotated atlas with clusters coloured according to their cell-type identity. **C** General library statistics of the whole dataset.
**D** Abstracted graph of lineage reconstruction (PAGA) showing most probable relationships between clusters. Each node corresponds to an annotated cluster.

The size of the nodes is representative of the number of cells in the cluster, the thickness of the edges is proportional to the connectivity probabilities. **E** Cell-type location in the organism accompanied by UMAP expression plot of marker gene for each cell type. **F** Expression plot and colorimetric RNA in situ hybridisation of *Chitinase2*, scale bar = 80 μm. **G** Expression plot and SABER-FISH for Avidin-like, scale bar = 50 μm. The white box shows a zoom on the tentacle base, scale bar = 10 μm.

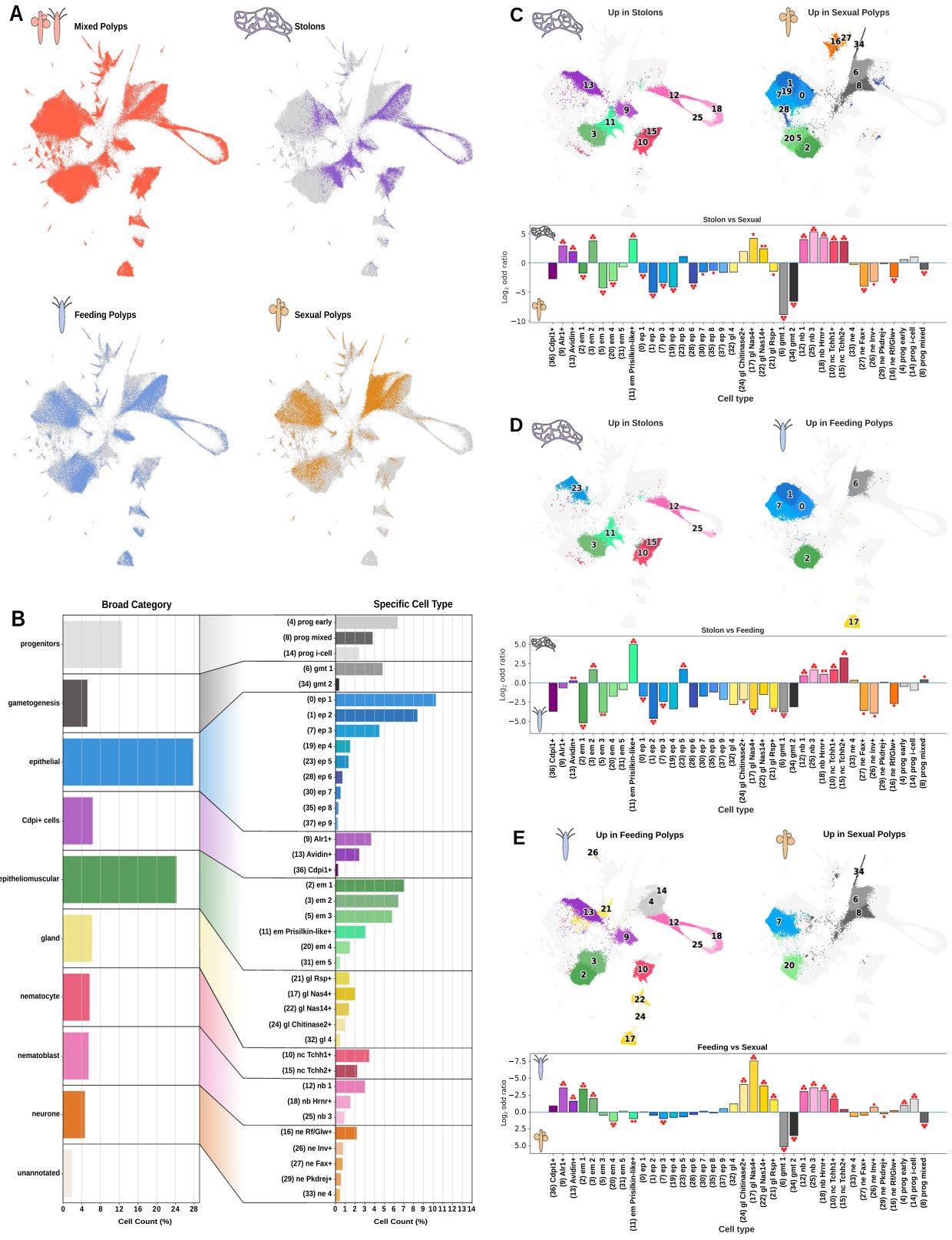

nematocytes and gland cells, which were prevalent in feeding polyps. To strengthen this finding we generated new analyses with the colony parts clustered individually (Supplementary Fig. 2), and annotated cells with the identity in the general analysis. This analysis corroborated our previous observations, which were consistent with our initial expectations. Interestingly, this analysis also reveals increased frequencies of individual cell clusters in the body parts. For instance,

*Conodipine*+ cells were more frequent in feeding polyps and stolons, and rare in sexual polyps. Furthermore, a cluster of *Prisilkin-like*+ cells was almost exclusive to stolons.

To statistically analyse these observations, we first calculated the aggregated cell counts for each broad cell-type category and individual cell type (Fig. 2B and Supplementary Data 7). We then calculated log2 odd ratios comparing feeding, sexual polyps and stolons

**Fig. 2 | Cell proportions of colony parts across the single-cell atlas. A** UMAPs showing presence of the different colony parts in the dataset (Mixed Polyps, Stolons, Feeding and Sexual Polyps). **B** Barplots showing the cell count percentage at the broad cell category and specific cell cluster levels. **C** Compositional comparison between stolon cells and cells coming from sexual polyps. On the top half area of each sub-item are the UMAPs of clusters enriched when the FDR was higher than 0.0005 in that respective condition ('up in' left or right). On the bottom half of each sub-item are the calculated logarithms of the odd ratios of enrichment output from the fisher test for each cluster. Each asterisk present on top of the barplot translates to a level of significance under different FDR (* = 0.05, **=0.005, ***=0.0005). **D** Compositional comparison between cells coming from stolon samples and feeding polyp cells. On the top half area of each sub-item are the UMAPs of clusters

enriched when the FDR was higher than 0.0005 in that respective condition ('up in' left or right). On the bottom half of each sub-item are the calculated logarithms of the odd ratios of enrichment output from the fisher test for each cluster. Each asterisk present on top of the barplot translates to a level of significance under different FDR (* = 0.05, **=0.005, ***=0.0005). **E** Compositional comparison between cells sampled from sexual and feeding polyps. On the top half area of each sub-item are the UMAPs of clusters enriched when the FDR was higher than 0.0005 in that respective condition ('up in' left or right). On the bottom half of each sub-item are the calculated logarithms of the odd ratios of enrichment output from the fisher test for each cluster. Each asterisk present on top of the barplot translates to a level of significance under different FDR (* = 0.05, **=0.005, ***=0.0005).

(Fig. 2C–E). Finally, we used scCODA, a Bayesian model, to analyse compositional changes in single-cell data (Fig. 2C–E) to detect credible changes. We leveraged distinct False Discovery Rates (FDRs, *=0.05,** = 0.005, *** = 0.0005) to gain an understanding of the different credible intervals within our results.

We first compared stolons to sexual polyps (Fig. 2C). Top credible effects (FDR 0.0005 = ***) with high log2 odd ratios in stolons included nematocytes (10, 15) and nematoblasts (12, 18, 25), consistent with a lower number of nematocytes in sexual polyps. These top credible effects also contained two epitheliomuscular clusters (3, 11) and two *Conodipine*+ clusters (9, 13), including *Alr1*+ cells and *Avidin*+ cells. Conversely, top credible effects in sexual polyps included the germ cell clusters (6, 34), a mixed progenitor cluster (8), a range of epithelial (0,1,7,19,28), epitheliomuscular (2, 5, 20), and neuronal (16, 27) cell clusters.

We then compared stolons to feeding polyps (Fig. 2D). Top credible effects with high log2 odd ratios in stolons included nematocytes (10, 15) and nematoblasts (12), indicating a higher frequency of nematocytes in stolons when compared to feeding polyps. These top credible effects also included an epithelial cluster (23) and two epitheliomuscular clusters (3, 11). Cluster 11 had the highest log2 odd ratio of this comparison, and was also highly enriched in stolons vs sexual polyps, indicating its high specificity to stolons. We termed this cluster *Prisilkin-like*+ cells due to the high expression of LOC130630016, which was annotated as a *Prisilkin-39-like* protein (Supplementary Data 3–5). In addition, another marker of this cluster wasLOC130630016, annotated as *Shematrin-like*. Interestingly, *Prisilkins* and *Shematrins* are glycine-rich repeat-containing proteins found in molluscan shells[48,49]. We then examined the top credible effects with high odd ratios in feeding polyps. These included mixed progenitors (6), epithelial clusters (0, 1, 7), and an epitheliomuscular cluster (2). These epithelial and epitheliomuscular clusters were also credibly enriched in sexual polyps vs stolons and therefore are highly specific of polyps.

Finally, we compared feeding to sexual polyps (Fig. 2E). Top credible effects with high log2 odd ratios in feeding polyps included a range of gland clusters (17, 21, 22, 24), nematocytes (10), nematoblasts (12, 18, 25), *Conodipine*+ cells (9, 13), and epitheliomuscular clusters (2, 3). Conversely, the top credible effects in sexual polyps included the germ cell cluster (6, 34), a mixed progenitor cluster (8), an epithelial cluster (7), and an epitheliomuscular cluster (20).

Altogether, our results revealed the cellular composition of the main *Hydractinia* colony part. Sexual polyps are made of epithelial and epitheliomuscular cell types, neurones, and germ cells, and lower in gland cells and nematocytes. Feeding polyps contain abundant epithelial, epitheliomuscular, nematocyte, gland, *Conodipine +*, and neuronal cells. Stolons possess fewer neuronal and gland cells but have abundant nematocytes and *Conodipine*+ cells. Finally, our data show that an epitheliomuscular cell type is highly specific to stolons. In summary, we find that colony parts seem to be defined by a combination of specific cell types and by different proportions of shared cell types.

## The transcriptional landscape of *Hydractinia* cell types

We then investigated the gene modules that underlie cell-type differentiation per cell type and colony part. We used Weighted Gene Coexpression Network Analysis (WGCNA)[50] to identify genes with correlated expression patterns. WGCNA calculates an Adjacency and Topological Overlap Matrix (TOM) that captures the interconnectedness and co-expression patterns among genes, providing insights into modular structures and regulatory relationships within biological networks. Unlike marker-finding algorithms, this approach can detect gene sets that are expressed in one single-cell type or coordinately expressed in several. We identified 7,547 genes distributed over 38 modules of gene co-expression, corresponding to individual cell clusters and broad groups (Fig. 3A, Supplementary Data 2, and Supplementary Fig. 3A, B). We used Gene Ontology (GO) analysis to extract biologically relevant terms for each cell type (Fig. 3B and Supplementary Data 8). Among broadly expressed gene modules we found a module expressed broadly in epithelial cells, enriched in GO terms such as "cell junction assembly" (Fig. 3B). We also found a module expressed broadly in epitheliomuscular cells, enriched in GO terms such as "smooth muscle cell differentiation" and "digestive tract morphogenesis" (Fig. 3B), consistent with the dual muscular and digestive epithelium nature of these cells. In addition, we found gene modules enriched in i-cells and mixed progenitors, enriched in GO terms related to DNA synthesis and replication as well as chromatin organisation (Fig. 3B). Interestingly, a module expressed in *Conodipine* + cells was enriched in biosynthetic and metabolic genes, consistent with a role in venom synthesis. Finally, we found a module expressed in *Prisilkin-like*+ cells, enriched in GO terms such as "chitin metabolic process". The genes annotated with these GOs include chitin synthase (LOC130613296) as well as chitinase (LOC130641097) and chitin deacetylase enzymes (LOC130614146), consistent with the chitinous nature of *Hydractinia* stolons[51–53]. Our data show that *Prisilkin-like*+ cells express chitin biosynthetic genes and are therefore key players in stolon chitinisation.

We then aimed to elucidate the similarities between cell-type expression modules to detect more subtle patterns of gene co-regulation between different cell types. One key advantage of WGCNA over marker-finding algorithms is that connections between modules can be derived from the structure of the underlying graph. To do this, we visualised the WGCNA TOM matrix as a network, and identified several graph-connected components that match the WGCNA modules (Fig. 3C). This analysis shows that connected components and WGCNA modules largely overlap, i.e. genes within a module are mostly connected amongst themselves but also have connections with other modules that reveal the similarities between these genetic programmes. To visualise this, we plotted a network where the nodes represent each gene module and the edges represent the number of co-expressed genes from different pairs of modules (Fig. 3D). This analysis recovered known relationships such as the connection between nematoblasts, nematocytes, and neurones[24,44,54–56], or the similarity between germ cells, mixed progenitors and

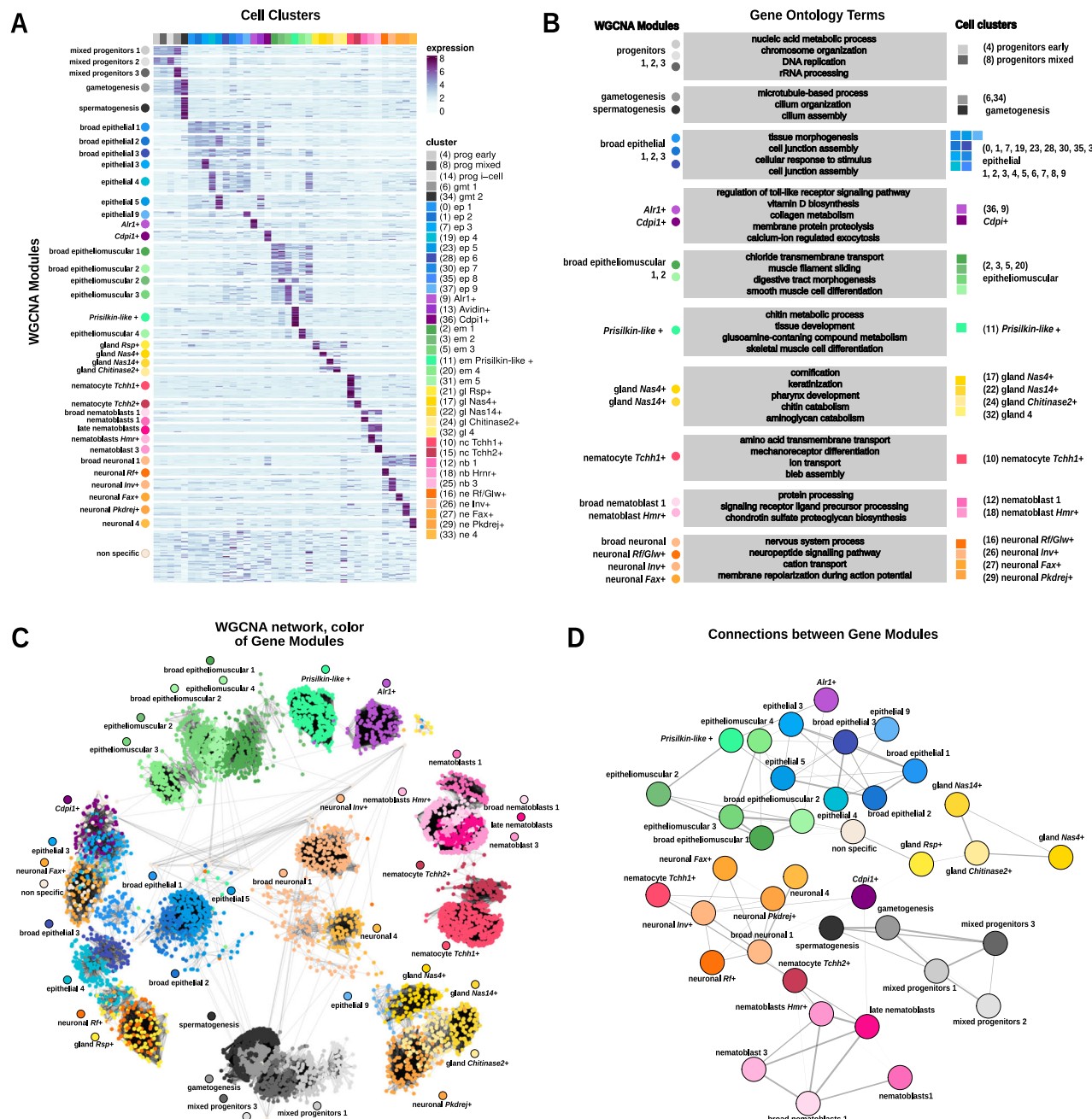

**Fig. 3 | Regulatory landscape of whole colonies in *Hydractinia*. A** Expression heatmap of 7547 genes over 38 modules of co-expression (rows and coloured circles) sorted by annotated cell type (columns and coloured squares). Colour intensity represents normalised expression. **B** Summarised gene ontology terms enriched in each module (left and coloured circles) associated with the respective cell cluster (right and coloured squares). **C** Network visualisation of WGCNA modules using the Kamida–Kawai layout algorithm. Each dot represents a gene and the respective colours are the cluster of the highest expression of a determined gene. Edges are a representation of co-expression values. Each module name has a circle with the corresponding colour placed above. **D** Module visualisation of the gene network using the large connected graphs layout algorithm, where co-expression values are summarised between different modules, showing the associations between them.

i-cells[57]. Furthermore, this analysis shows that *Conodipine*+ cells have mixed patterns, with the *Cdpi1*+ module connected to neurones, and the *Alr1*+ module connected to both epithelial and epitheliomuscular cell modules. Importantly, this analysis showed that the stolon-enriched *Prisilkin-like*+ module was connected to other epitheliomuscular clusters, corroborating their epitheliomuscular identity, but also revealing its connection with the *Alr1*+ module. These results show that these cell types have genes that

are co-expressed and suggest a functional or spatial relationship between them. Altogether, these analyses reveal the transcriptional patterns of each *Hydractinia* cell type and their similarities, providing a foundation for future functional studies.

**Transcription factor regulatory profiles in *Hydractinia***
We then aimed to identify TFs that may be driving the differentiation of *Hydractinia* cell-type diversity. We annotated 801 TFs and identified a

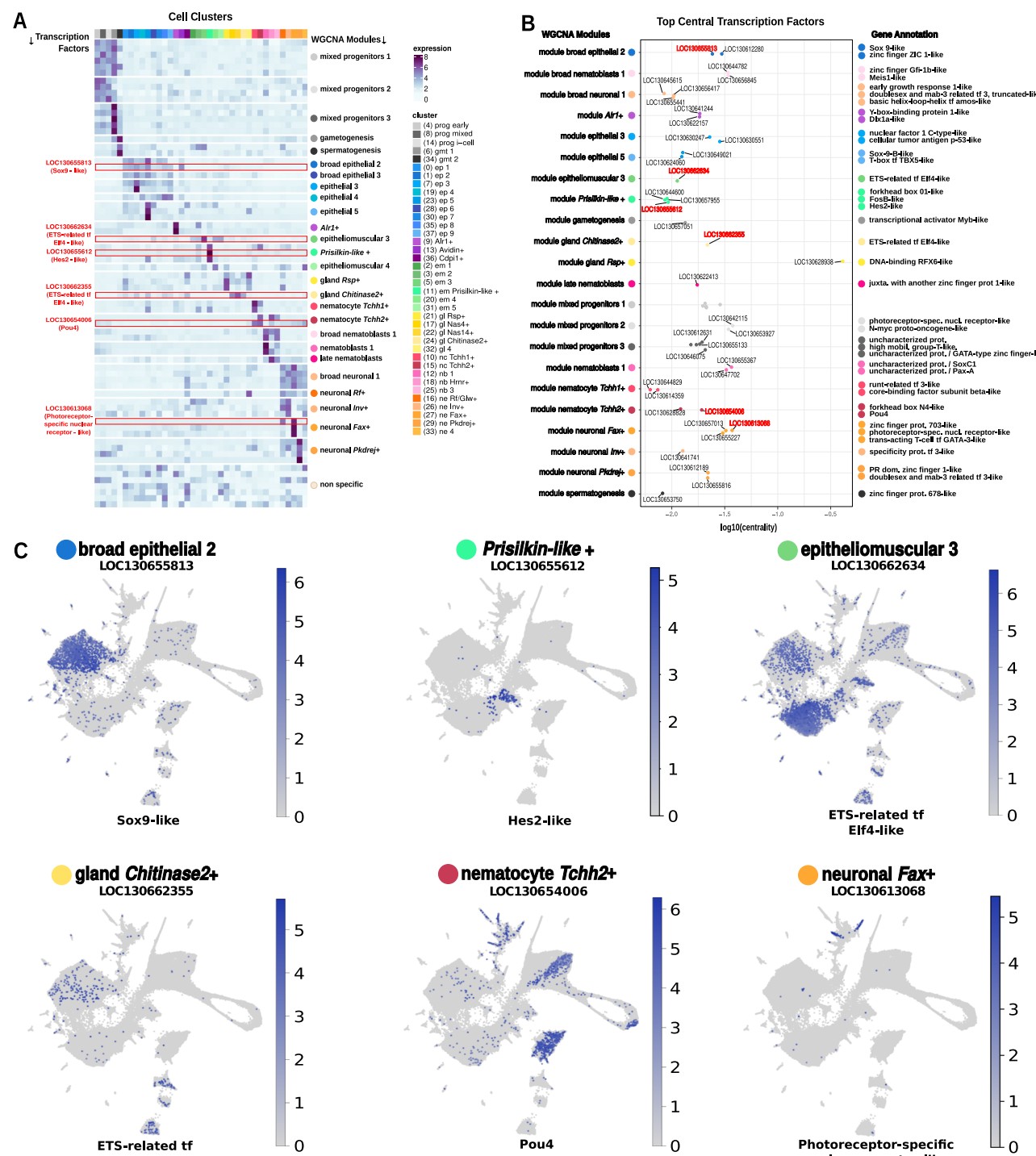

**Fig. 4 | Transcription factor expression landscape across whole colonies of *Hydractinia*. A** Expression heatmap of the transcription factors found in the output of WGCNA over 38 modules of co-expression (rows and coloured circles) sorted by annotated cell type (columns and coloured squares), highlighting in red the factors selected shown in (**C**). **B** Stripplot graphing the obtained top central transcription factors in WGCNA modules (coloured circles) with their respective annotations, highlighting the selected factors in red. **C** Feature expression plots of selected top central transcription factors indicating which WGCNA module does each belong to.

set of 69 of them with cell-type-specific expression (Fig. 4A and Supplementary Data 2). We then exploited our WGCNA TOM graph approach to obtain centrality values for each TF in each module. In essence, TFs with a high centrality within a module are connected to many genes within the module and therefore are highly correlated with them and candidate regulators. We plotted the top central TFs for each module (Fig. 4B, C), which included several TFs previously described. Among the top central TFs we found Sox9-like (broad epithelial), ETS-related (epitheliomuscular and gland cells), Forkhead box protein N4-like (nematoblast and nematocytes), and Photoreceptor-specific nuclear receptor-like (neurones).

One transcription factor identified as top central, to the 26_nematocyte_Tchh2+ module, was a POU domain, class 4 transcription factor (Supplementary Fig. 4 and Supplementary Data 9). In mammals, POU4 is vital to proper development of mechanosensory cells and neurones in the inner ear and surrounding structures[58].

POU4's role in the development of mechanosensory neurones is consistent across metazoans[59]. In *Nematostella*, NvPOU4 RNA in situ hybridisation predominantly marks late differentiating stinging cells (nematocytes) at the distal end of the tentacles, along with some differentiating neurones in the body column. NvPOU4−/− homozygous knockout mutants fail to generate terminally differentiated nematocytes[60]. They produce nematoblasts devoid of a capsule or stinging apparatus (harpoon), but that are still marked by NvNCol3 (a marker of early nematogenesis). NvPOU4 has been proposed to have an ancestral function in mechanosensory cell development and appears in cnidarians to have a dual role, acting also in the production of mechanosensory machinery in the cnidarian-specific stinging cell type, cnidocytes[60,61]. The UMAP expression plot of *Hydractinia Pou4* is consistent with this dual role, with expression in both the neural and nematocyte lineages (Fig. 4C). The co-expression of *Pou4* and *Trichohyalin* in the *Tchh1* and *2*+ nematocyte types suggests a potential co-opting of gene modules traditionally associated with mechanosensory cells and follicle development for the production of the unique mechanosensory machinery found in nematocysts.

Our analysis also unveiled TFs that are expressed in other cell types characterised in this study. For instance, top central factors in the *Prisilkin-like*+ module include a Forkhead box protein O1-like, a FosB-like and a Hes2-like TFs that are highly specific to this stolon-enriched cell type. Collectively, these results reveal the TFs that are specific to *Hydractinia* cell types and will open numerous research avenues to investigate their role in cell-type differentiation and evolution.

### *Alr1*+ cells in epidermal tissues of polyps and stolons express allorecognition genes

The ability to discriminate self from conspecific non-self (allorecognition) has evolved multiple times in animals[62]. Vertebrate allorecognition is thought to be a side effect of T-cell/MHC-mediated immunity, given that these animals do not naturally encounter allogeneic cells[63]. In colonial invertebrates that do encounter allogeneic tissues during growth, allorecognition systems mediate stem/germ cell parasitism, preventing unrelated allogeneic stem/germ cells from invading their tissues and contribute to somatic cells and gametes[12]. *Hydractinia* is one of only a few invertebrates in which the molecular mechanism of allorecognition has been elucidated. It consists of at least two genes—*Alr1* and *Alr2*—that reside on a single genomic region called the allorecognition complex (ARC) and are highly polymorphic. Shared alleles at the ARC allow allogeneic fusion while rejection occurs when no alleles are shared[64]. *Alr1* and *Alr2* are thought to encode homophilic cell adhesion molecules. It has been shown that these proteins bind only to their own or to nearly identical proteins[65]. The structure of the ARC has been studied previously but these studies could not fully resolve the genomic context of allorecognition genes[66]. Moreover, the expression pattern of allorecognition genes has not been shown.

Using a new, chromosome-level genome assembly[18], we have identified the location of the allorecognition complex on Chromosome 3. We find a total of 40 Alr/Alr-like genes distributed across chromosome 3, including some potential pseudogenes (Fig. 5A). Two distinct gene clusters are present containing canonical Alr1 and Alr2, respectively. Amongst the 40 genes identified within the ARC, some show homology to either Alr1 or Alr2, while others were returned as hits with similar confidence when using either gene as a query, highlighting the high similarity across the Alr gene family. When querying the entire genome, we identified 3 putative Alr homologues outside of the ARC region (LOC130636598, LOC130644743 and LOC130647443) present at the end of Chr3 (outside of the ARC) and on Chr5 and Chr6, respectively (Supplementary Data 10). *Alr1* expression was highly specific to cluster 9 (Fig. 5B) while *Alr2* had a broader expression pattern in multiple clusters (Supplementary Fig. 5A). Other Alr-related

gene expression was also specific to cluster 9 in varying levels (Supplementary Fig. 5B).

We generated antibodies directed against Alr1 (Supplementary Fig. 6 and Supplementary Data 11) and studied its spatial distribution. We found Alr1 protein in epidermal epithelial cells in the polyp body column (Fig. 5C) and in stolons, with particularly high levels in stolonal growing tips (Fig. 5D). This is consistent with the known colony parts that exhibit allorecognition in the animal[53,67,68].

To functionally study which tissues mediate allorecognition, we performed pairwise polyp-polyp grafts using both isogeneic polyp pairs and allogeneic, presumably *Alr1/Alr2*−mismatching ones. Polyps were removed from their colonies by a transverse cut close to the polyp-stolon boundary. One polyp was briefly incubated in a black ink solution to visualise the interface. The polyps were then strung on a glass capillary, wound side appressed, forcing contact between allogeneic epidermis and gastrodermis (Fig. 5E). The contact area was studied histologically at 4 and 12 h post grafting. We found that in isogeneic colonies, both epidermis and gastrodermis of the grafts fused, resulting in tissue continuity between the polyps (Fig. 5F). In allogeneic, *Alr1/Alr2*−mismatching pairs, by contrast, the gastrodermis fused indiscriminately but the epidermis remained unfused. Eventually, the unfused epidermis of both polyps contracted, separating the fused gastrodermis, resulting in the parting of the two grafts (Fig. 5G). These experiments are consistent with the localisation of Alr1 in the epidermis. Moreover, it fits the mode of natural allogeneic encounters in *Hydractinia* where only the epidermis but not the gastrodermis establishes direct contacts with allogeneic counterparts. Finally, *Hydractinia*'s pluripotent cells–the i-cells–are exclusively present in the interstitial spaces of the epidermis such that even unlikely contact between allogeneic gastrodermal tissues would not result in stem/germ cell parasitism.

### *Prisilkin-like*+ cells in stolons at the base of polyps express biomineralization genes

We then aimed to further investigate the nature of *Prisilkin-like*+ cells with a focus on genes which appeared either as marker genes (Supplementary Data 3 and 4) or in the WGCNA gene module which was expressed in these cells (Supplementary Data 2). Our previous results show that these epitheliomuscular cells are highly enriched in stolons (Fig. 2), expressing genes similar to the molluscan shell matrix protein-coding genes *Shematrin* and *Prisilkin* (Supplementary Data 3 and 4) as well as chitin synthesis genes (Fig. 3). To gain further insight into *Shematrin*-like and *Prisilkin-like* genes, we analysed their genomic locations, taking advantage of the newly assembled chromosome-scale *Hydractinia* genome[18]. Interestingly, both genes annotated as *Shematrin* and *Prisilkin* were located in the same genomic region in chromosome 2 (Fig. 6A). We observed that several other genes surrounding this genomic location also had similar repeats, indicating the presence of a genomic cluster likely originated by tandem duplications (Fig. 6A). Most of these genes had substantial specific expression in the *Prisilkin-like*+ cell cluster (Fig. 6B).

To understand their sequence similarities, we performed a series of preliminary BLAST analyses which failed to provide robust results due to the highly repetitive nature of these sequences. We also aimed to obtain gene alignments of the *Hydractinia Prisilkin-like*+ cluster genes and other genes annotated as *Shematrins*, which resulted in poor alignment and support values. Finally, to structurally characterise these genes we devised a repeat finder algorithm, and used SignalP 6.0[69] to predict signal peptides. Our repeat finding algorithm searches for repeats with a $XG_nX$ (where X represents amino acids I, L, V, Y) structure where n can have a value between 1 and 4 (singlets, duplets, triplets, quadruplets), typical of molluscan *Shematrins*[49]. *Hydractinia Prisilkin-like*+ cluster genes all had a majority of singlet and duplet repeats, and 7 out of 9 had a predicted signal peptide (Fig. 6C). We found another gene cluster in a nearby genomic location containing 8

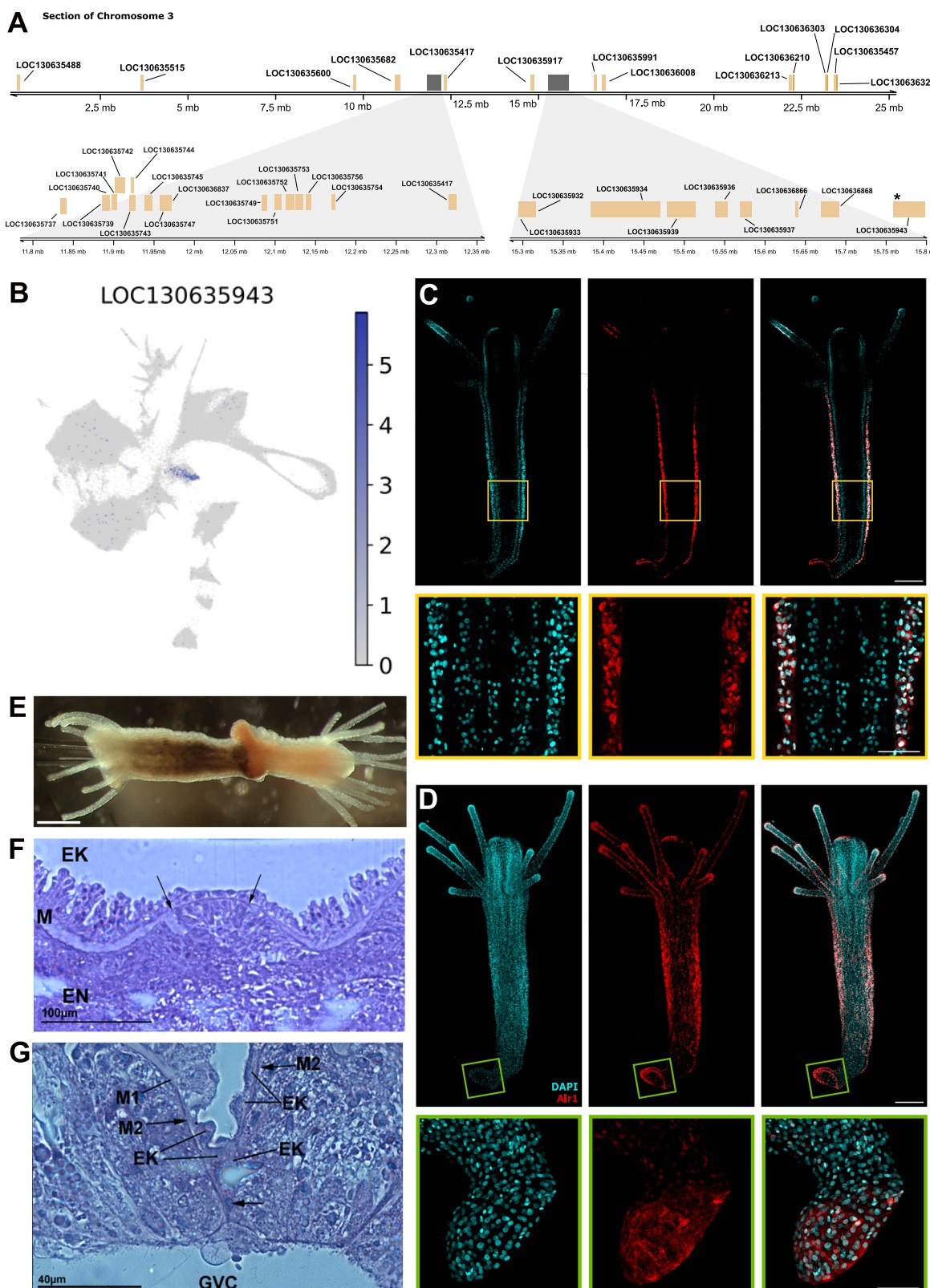

**Fig. 5 | The Allorecognition Complex and its expression. A** Region on chromosome 3 of Hydractinia depicting the Allorecognition Complex (ARC) including Alr1 and Alr-like genes. **B** UMAP expression plot of LOC130635943, an Alr1 gene. **C** Single confocal slice of Alr1 immunofluorescence in a feeding polyp, showing Alr1 localisation in the epidermis. Scale bar = 80 μm. Zoom is highlighted in yellow. Scale bar = 30 μm. **D** Maximum projection of a confocal stack of Alr1 immunofluorescence in a feeding polyp. Scale bar = 80 μm. Zoom highlighted in green shows a growing stolon. Scale bar = 40 μm. **E** Two feeding polyps grafted at the aboral end. Scale bar = 500 μm. **F** Histological section of the interface between two isogeneic polyps, showing tissue continuity of epidermis and gastrodermis. Arrows point to the new mesoglea. M old mesoglea, EK epidermis, EN gastrodermis. **G** Histological section of the interface of two incompatible, allogeneic polyps. M1 old mesoglea, M2 new mesoglea, GVC gastrovascular cavity.

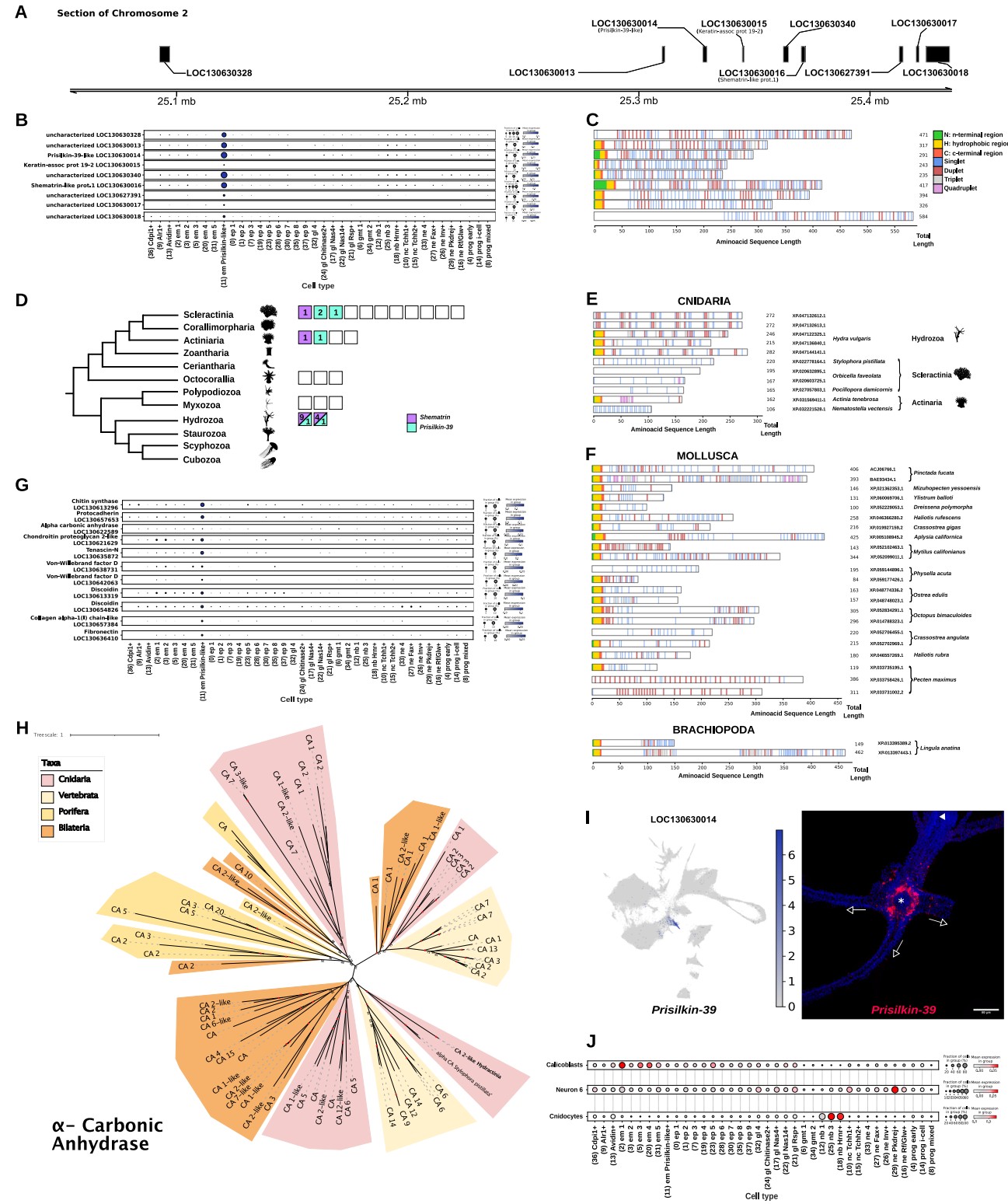

further repeat-containing genes with a very similar structure in chromosome 2 and two more genes very close together in chromosome 3 (Supplementary Fig. 7).

We then wondered if other *Shematrin*-like genes are present in other cnidarian species. Since these are difficult to identify by BLAST algorithms, but are sometimes annotated as "*Prisilkin*" and "*Shematrin*" in NCBI annotated genomes, we searched currently available cnidarian genomes for these annotations, and grouped them phylogenetically[70]. This analysis uncovered annotated *Shematrins* or

*Prisilkins* in 7 out of 23 species with annotated genomes (Fig. 6D). *Hydra vulgaris*, *Stylophora pistillata*, *Orbicella faveolata*, *Pocillopora damicornis*, *Actinia tenebrosa* and *Nematostella vectensis* have *Shematrin* genes with similar repeats (Fig. 6E).

To compare these results to molluscan *Shematrin* genes, we detected repeats using the same algorithm in 22 molluscan *Shematrins* (Fig. 6F). We included 2 *Shematrins* from Brachiopoda for reference (Fig. 6F). This analysis revealed that their structure is very similar, with a signal peptide followed by repeats with a singlet and duplet majority.

**Fig. 6 | Architecture and phylogenetic relationships of genes primarily expressed in *Prisilkin-like*+ cells. A** Region on chromosome 2 of *Hydractinia* depicting genomic location of *Prisilkin-39* and one annotated *Shematrin-like* gene. **B** Dot plot showing individual expression levels of genes specific to the *Prisilkin-like*+ cluster in the same genomic location ranges of (**A**). **C** Signal peptide regions (N, H, and C-terminal regions) and glycine repetitions (singlet, duplet, triplet, quadruplet) across complete amino acid sequences of all genes in (**B**). (tomato red; C, lime green; N, gold; H, cornflower blue; singlets, indian red; duplets, light grey; triplets, plum; quadruplets). **D** Phylogeny of cnidaria adapted from De Biasse et al. depicting number of annotated proteomes in NCBI (empty blocks), number of annotated *Shematrin-like* (Lilac blocks) and *Prisilkin-39* proteins (Turquoise blocks). Clade silhouettes were obtained from www.phylopic.org and are available under public domain, except for the Octocorallia silhouette (credit to Qiang Ou, CC-BY 3.0 https://creativecommons.org/licenses/by/3.0/). **E** Signal peptide regions (N, H, and C-terminal regions) and glycine repetitions (singlet, duplet, triplet, quadruplet) across complete amino acid sequences of all *Shematrin-like* and

*Prisilkin-39-like* annotated proteins in cnidarians. **F** Signal peptide regions (N, H and C-terminal regions) and glycine repetitions (singlet, duplet, triplet, quadruplet) across complete amino acid sequences of a selection of *Shematrin-like* and *Prisilkin-39-like* annotated proteins in molluscs. **G** Dotplots showing individual expression levels of genes found *Hydractinia* related to biomineralization pathways. **H** Unrooted phylogenies of genes annotated as Alpha Carbonic Anhydrase. In each tree, animal taxa are highlighted with legend provided accordingly. All sequences belonging to *Hydractinia symbiologicarpus* are highlighted in bold. Leaves on trees include species names, accession numbers if obtained from NCBI, and sequence annotation. Ultrafast bootstrap values are reported on nodes with red dots if > = 95. **I** Maximum projection from a confocal stack of Prisilkin-39 SABER-FISH in a 3-day post metamorphosis, substrate adhered, feeding polyp, imaged from the aboral pole. *Indicates oral pole, arrows indicate direction of outgrowth of stolon, scale bar = 80 μm. **J** Selection of dotplots showing scored expression of one-to-one ortholog genes present in annotated clusters of *Stylophora pistillata* with minimum fold change of 1 (Calicoblasts, Neurone 6 and Cnidocytes).

Altogether, these analyses show that *Hydractinia* stolon-enriched *Prisilkin-like*+ cells express a group of clustered *Shematrin-like* and *Prisilkin-like* genes similar to molluscan *Shematrins*. These genes are present in other related cnidarians but their wider presence in the cnidarian phylum cannot be identified with current annotation methods.

As *Shematrins* have been characterised as molluscan shell proteins and are involved in biomineralization in other organisms too, we searched for the expression of other bona-fide biomineralization genes. Among *Prisilkin-like*+ cell markers and WGCNA module genes, we identified 11 genes that may be involved in biomineralization (Fig. 6G). Among these, we found homologues of carbonic anhydrases (Fig. 6H and Supplementary Data 12), chitin synthases and protocadherins (Supplementary Data 12). We confirmed that these *Hydractinia* genes are homologous to related genes in other animal groups by phylogenomic analyses (Supplementary Data 13). These analyses confirmed that *Hydractinia Prisilkin-like*+ cells express a biomineralization gene programme. To gain insights into the localisation and function of *Prisilkin-like*+ cells in *Hydractinia*, we designed probes against LOC130630014, annotated as *Prisilkin-39* and performed in situ hybridisation experiments (Fig. 6I). We found that *Prisilkin-like*+ cells are abundantly present in the stolon, and accumulated in the section that intersects with the base of polyps. This suggests a model where *Hydractinia* forms attachment points to the substrate (naturally, hermit crab shells) via the *Prisilkin-like*+ cells, a stolon-enriched cell type that expresses biomineralization genes, including genes akin to genes involved in molluscan shell formation as well as chitin synthesis genes. Similar cell types and genes have not been found in other cnidarians, raising the possibility that they are a colonial cnidarian innovation.

Finally, we aimed to elucidate the evolutionary relationship between *Prisilkin-like*+ cells and other cnidarian types involved in biomineralization. Recently, the cell-type atlas of the stony coral *Stylophora pistillata* has revealed that these corals have calicoblasts, a type of epidermal cells that express carbonic anhydrases among other enzymes[71]. To elucidate if *Stylophora* calicoblasts are similar at the transcriptomic level to *Hydractinia Prisilkin-like*+ cells we obtained the top markers of calicoblasts, identified one-to-one orthologs in *Hydractinia*, and scored their expression in our single-cell dataset. These scores were high in several epitheliomuscular and epithelial cell types, but not in *Prisilkin-like*+ cells (Fig. 6J). As a comparison, we performed the same analyses using markers of *Stylophora* neuronal types and cnidocytes. These scores were indeed high in *Hydractinia* neuronal types and nematoblasts, evidencing the transcriptomic similarity of these types, likely originated by an evolutionary homology relationship. In contrast, calicoblasts and *Prisilkin-like*+ cells are not transcriptionally similar and therefore have likely originated as two independent cell-type innovation events in the evolution of the *Stylophora* and *Hydractinia* lineages.

## Discussion

Our *Hydractinia* cell atlas reveals the cellular composition of distinct colony parts of a colonial cnidarian. Previous cnidarian cell atlases have profiled solitary species such as *Hydra* and *Nematostella* or considered colonial cnidarians as a whole[23–25,27–29,47,72,73]. We show that most cell types are present in all parts of the colony, albeit in different proportions, with fewer cell types being part-specific (Fig. 7). Therefore, it appears that a cell 'mix-and-match' approach has been the major mechanism in the evolution of new colony parts in *Hydractinia*. In a broader sense, using different combinations of a similar cellular repertoire may have facilitated the multiple gains and losses of characters, such as medusa stage and coloniality, in hydrozoan evolution[2,74] and could be a common mechanism of colony evolution across Metazoa. We demonstrate that single-cell methods are ideal for studying cellular composition in colonial animals. Future studies will apply these methods to other colonial animals, such as colonial ascidians and bryozoans.

We identified 53 cell clusters, 38 of which we annotated as specific cell types, and 15 of which remain unannotated. We validated the 38 cell-type annotations through several in silico methodologies. The atlas shows known cell types that are found in several single-cell atlases, such as nematocytes, epithelia, and neurones as well as multiple *Hydractinia*-specific novel cell types. We analysed gene sets with expression signatures that can be correlated mostly one-to-one to the obtained cell types and used them to define regulatory profiles from 38 gene co-expression modules and transcription factors. This fine-grained characterisation of signalling pathways controlling colony homoeostatic processes, allows us to understand better molecular mechanisms and gene regulatory networks underlying evolutionary patterns of colony plasticity.

*Hydractinia* i-cells have been studied for over 140 years[16,67], being the first studied animal stem cells. Later studies predicted that i-cells are pluripotent, able to contribute to all somatic cells and gametes[5,75,76]. This ability of i-cells has been confirmed experimentally at single-cell resolution only recently[11]. Our cell atlas and the PAGA analysis computationally confirm what has been known experimentally. The congruence between the two approaches provides confidence to our results and places this model organism and its cell atlas in a unique position to study pluripotency and lineage commitment in vivo in a fully genetically tractable system. The PAGA analysis recapitulated the *Hydractinia* cell lineages' derivation from i-cells; however, Cluster 30, a sub-cluster of epithelial cells, appears to derive directly from i-cells rather than from the larger epithelial cluster as would be expected. This could be explained by the fact that most polyps from which the atlas was generated were fully grown. Since epithelial turnover in *Hydractinia* is slow[51], committed epithelial progenitors would be rare in our sample, making their precise position in the PAGA cladogram difficult to determine.

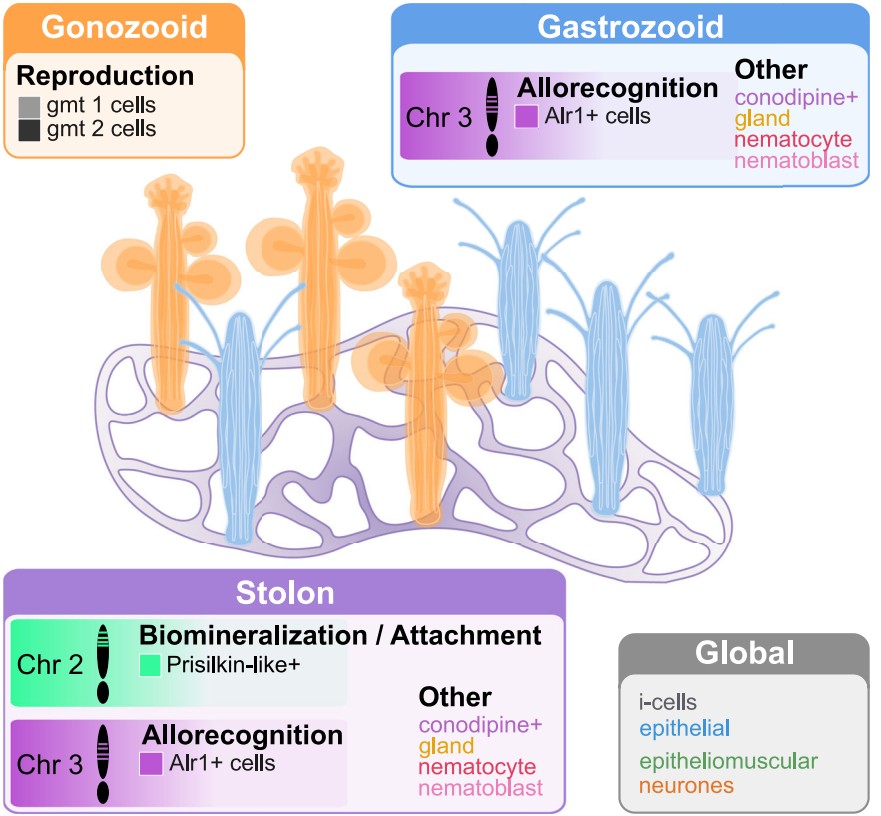

**Fig. 7 | Summary of the relative cellular composition of different colony parts.** Reproductive polyps (gonozooids) are enriched in germ cells. Feeding polyps (gastrozooids) are enriched in cells that contribute to allorecognition and stinging relative to gonozooids. Stolons are similar to feeding polyps with respect to allorecognition and stinging but are also enriched in cells contributing to biomineralization and attachment to the substratum. i-cells, epithelial cells, epitheliomuscular cells, and neurones have similar proportions in all colony parts.

Cnidarian nematocytes are a unique cell type across the metazoan tree but appear to have co-opted gene programmes and transcription factors used for specification of mechanosensory cells in other organisms. Previous work in *Nematostella* highlighted a role for POU4 in the terminal differentiation of nematocytes[60]. Here, we demonstrate nematocyte-specific expression of a *Pou4* homologue, along with co-expression of *Trichohyalin-1* and *-2*, known markers of hair follicle development in other organisms[77] and venom genes. As such, it appears that nematocytes, although a unique cell type, do not rely on a unique gene programme for their function, rather that they employ a unique combination of common gene programmes, shared with other organisms.

Our analysis reveals new cnidarian cell types, such as the *Conodipine*+ cells, including *Avidin*+ cells and *Alr1*+ cells, that share many transcriptomic signatures with nematocytes, nematoblast, neurones, and epithelial cells. These cells express genes encoding for venom such as membrane protein proteolysis, and at least one homologue of the Membrane Attack Complex (MAC) and perforin family, a group of proteins studied before for being recruited into venom-injecting cells in cnidarians[78]. However, they lack the nematocyst apparatus. Venom-expressing cells that are not nematocytes are also present in *Nematostella*[79]. Further studies are required to fully confirm the identity and function of this broad cell type in *Hydractinia* colonies.

The *Conodipine*+ cell cluster 9 was also marked by *Alr1*, an allorecognition determinant gene that is physically located in the ARC on Chromosome 3. Our data show that Alr1 protein localises to epidermal epithelial cells that are the ones that naturally encounter allogeneic cells. Consistently, we find that the epidermal tissue is the one that actually performs the recognition. Since most allogeneic interactions between *Alr1-mismatching* colonies result in aggressive rejection, it would be interesting to investigate the role of Conodipine in the rejection response that follows non-self-recognition. *Hydractinia* is one of only two invertebrates (and the only cnidarian) in which the genetics of allorecognition has been studied in detail. Our results facilitate and encourage new studies on the evolution and mechanisms of self/non-self discrimination systems in metazoans.

*Hydractinia* and other closely related cnidarians have adapted to form colonies on molluscan shells inhabited by hermit crabs. *Hydractinia* colonies possess chitinous stolons[51–53] and form a crystalline mat with calcium carbonate[80], though the cellular and molecular basis for this process was previously unknown. Our data show that *Hydractinia* has *Prisilkin-like*+ cells, a cell type in their stolons that expresses enzymes for both chitin synthesis and biomineralization. They localise to the stolonal sections connected to the polyps, suggesting that these serve as attachment points. This cell type is not transcriptionally similar to the recently described calicoblasts of stony corals[81], suggesting that they may have arisen independently in the evolution of Hydrozoa and Hexacorallia.

Cnidarian skeletons have been morphologically classified as calcareous, corneous, or coriaceous[82], with an additional type, bilayered, described later[83]. Our data reveal that *Hydractinia Prisilkin-like*+ cells are responsible for chitin and biomineralization, likely forming the basis for coriaceous biomineralization. Stony corals, which have calcareous skeletons, possess calicoblasts that are likely not homologous to *Prisilkin-like*+ cells. Along with the recent finding that biomineralization in stony corals and octocorals arose independently[84,85], our data suggest a scenario where biomineralization evolved independently in several cnidarian lineages. Therefore, we propose that cell-type innovation happens at a low frequency when compared to the innovation of colony parts using preexisting cell types. Future

single-cell studies in closely related species will test this hypothesis. Nonetheless, cell-type innovation might have played important roles in the evolution of coloniality in cnidarians. In the future, single-cell techniques will enable the identification of additional biomineralizing cell types and other cell-type innovations, allowing further investigation into the cellular and molecular basis of adaptation.

## Methods

### *Hydractinia* culture and experimental manipulation

*Hydractinia symbiolongicarpus* male clone 291-10 was cultured as previously described[15]. Polyp samples were obtained by cutting them with fine surgical scissors close to the polyp-stolon boundary, with samples being taken from across the colony favouring fully developed polyps. Feeding and sexual polyps are easily distinguishable by morphology. Stolonal tissue was harvested by cutting away all polyps from a colony, followed by scraping the stolons with a single-sided razor blade.

### ACME dissociation

After sample collection, isolated colony parts were simultaneously fixed and dissociated using ACME[33] to generate distinct libraries per colony part. For libraries 09 (mixed feeding and sexual polyps) and 20 (mixed feeding and sexual polyps, and stolon-only sublibraries), ACME-a solution was prepared, on ice, using a 12:3:2:2:1 ratio of DEPC treated water, methanol, glacial acetic acid, glycerol and 7.5% N-Acetyl-Cysteine (Sigma A9165) in DEPC water. Dissociation was performed by placing ±100 polyps, in 30 ppt filtered seawater (FSW), into nuclease-free 1.5-mL tubes on ice. FSW was removed, followed by direct addition of 1 mL of ACME-a solution. Immediate vigorous pipetting was performed at the surface of the liquid to generate small bubbles (using a p1000 pipette, set to 700 μL), samples are completely dissociated after around 15 min of pipetting. After dissociation, samples in ACME-a were filtered through 40 μm Flowmi filters (Bel-Art™ SP Scienceware™ Flowmi™ Cell Strainers for 1000 μL Pipette Tips).

For libraries 27 (feeding polyps only) and 29 (sexual polyps only) ACME-b solution was prepared, on ice, using a 13:3:2:2 ratio of commercially sourced nuclease-free water, methanol, glacial acetic acid and glycerol. Prior to dissociation, ±100 polyps were placed in 1.5 mL lobind tubes in FSW. FSW was removed and replaced with anaesthetic solution (filtered 4% $MgCl_2$ made in 50:50 MilliQ $H_2O$ and FSW 30 ppt) and samples were left on ice for 15 min to anaesthetise. For removal of the glycocalyx and preservation of RNA, the anaesthetic solution was removed and replaced by 200 μL of 7.5% N-Acetyl-Cysteine (in anaesthetic solution), which was immediately removed and replaced by 1 mL ACME-b solution. Samples were dissociated as above, followed by filtration through 100 and then 40-μm PluriSelect mini cell strainers (SKU 43-10040-40).

After filtration, samples were centrifuged for 5 min at 1200× $g$ in a 4 °C cooled swing-out bucket centrifuge. The supernatant was aspirated and the pellet resuspended in 1× PBS in nuclease-free water (volume dependant on the number of sub-samples to be concatenated e.g., 200 μL for concatenation of 5 samples, for a final volume of 1 mL). After resuspension, samples were concatenated for a final volume of 1 mL, followed again by 5 min centrifugation at 1200× $g$ in a 4 °C cooled swing-out bucket centrifuge. The supernatant was aspirated and the pellet resuspended in 900 μL 1× PBS in nuclease-free water, followed by the addition of 100 μL DMSO (Sigma D8418). Tubes were gently inverted to homogenise the sample and DMSO, followed by storage at −80 °C for up to 3 months.

### Flow cytometry

Cells were prepared for SPLiT-seq by thawing on ice then centrifuging for 5 min at 1200× $g$ in a 4 °C cooled swing-out bucket centrifuge. The supernatant was aspirated and the pellet resuspended in 500 μL of 1% BSA in PBS followed by another round of centrifugation. The supernatant was aspirated and the pellet resuspended in 450 μL of 1% BSA in

PBS followed by filtration through a 50-μm CellTrics strainer (Sysmex CS678337 and 04-004-2327). In all, 50 μL of cell suspension was aliquoted and diluted by the addition of 100 μL 1% BSA in PBS for a final 1 in 3 suspension. The 1 in 3 suspension was stained with 1.5 μL of DRAQ5 (1 in 10 dilution of a 5 mM stock, eBioscience, Invitrogen 65-0880-92) and 0.6 μL of Concanavalin-A (Con-A) conjugated with Alexa Fluor 488 (1 mg/mL stock, Invitrogen C11252) and incubated in the dark at room temperature for 25 min.

The stained 1 in 3 cell solution was assessed by flow cytometry (CytoFlex S Flow Cytometer, Beckman Coulter) and the number of singlet (based on FSC-H vs FSC-A), nucleated (based on DRAQ5 vs FSC-A, gated as a sub-population of singlets) and intact (Con-A positive vs FSC-A, gated as a sub-population of nucleated) was measured per 10 μL, in triplicate. An average of the three readings for each sample was used to calculate the number of cells per microlitre in the original cell suspension.

### SPLiT-seq barcoding

SPLiT-Seq barcoding was performed as described[33] with several modifications. All primers and barcodes were provided by IDT. Cells from libraries 09 and 20 were diluted in 0.5× PBS for a final concentration of 625 cells/μL and 8 μL of this suspension was added to each well of the RT/R1 barcoding plate (for a final concentration of 5000 cells per well) with 5× RT Buffer (Thermo Scientific, EP0753), Maxima H Minus RT (Thermo Scientific, EP0753), SUPERase-In RNAse inhibitor (20 U/μL, Invitrogen, AM2696) and 10 mM/each dNTPs (NEB, N0447S). For libraries 27 and 29 cells were diluted in 0.5× PBS to a final concentration of 770 cells/uL and 6.5 μL of this suspension was added to each well of the RT/R1 barcoding plate (for a final concentration of 5000 cells per well) with NEBuffer r3.1 (NEB, B6003S), T4 Ligase Buffer 10× (NEB, M0202L), T4 DNA ligase (400 U/μL, NEB, M0202L). The addition of 6.5 μL of cell suspension, rather than 8 μL, in libraries 27 and 29 was to account for a modification to the RT master mix in these libraries. The RT master mix for libraries 27 and 29 contained 10% w/v PEG8000, to help with pellet aggregation in later centrifugation steps, thus increasing the volume of the master mix to 9.5 μL per well in the RT/R1 plate. All further steps of barcoding, sorting, lysis, cDNA purification, template switching, PCR and qPCR amplification and size selection were performed as per the original protocol.

Tagmentation and round 4 barcoding was performed in the same way for all libraries. The concentration and average fragment size of the size selected libraries was assessed by Qubit dsDNA HS assay (Invitrogen Q33230) and Bioanalyzer High Sensitivity DNA Analysis (Agilent Technologies AGLS5067-4626), respectively. The Nextera XT DNA Library Preparation Kit (Illumina 15032354) was used to perform tagmentation of 1 ng of cDNA, as calculated previously, per the manufacturer's guidelines. The final Round 4 barcode was added by PCR by the addition of 20–22 μL of tagmented cDNA. In all, 15 μL of Nextera XT Kit PCR mix, 1 μL of 100 μM Tagmentation Master Primer (BC_0018) and 1 μL of 100 μM Round 4 Barcode (one of BC_0076-BC_0083 per sub-library) into a PCR tube. The reaction was incubated in a thermocycler at 72 °C for 3 min, 95 °C for 30 s, then 13 cycles of 95 °C for 10 s, 55 °C for 30 s and 72 °C for 30 s, followed by 72 °C for 5 min and 4 °C hold.

The resultant libraries were again size selected at 0.7 and 0.6× as per the original protocol and assessed by Qubit and Bioanalyzer for pooling in an equimolar mix for sequencing. Pooled libraries were sequenced both shallow (10 M reads per 10k cells) and deep (133 M reads per 10k cells).

### Diamond blast annotation

We implemented diamond v2.0.8.146[35] to corroborate orthologs present in our reference genome on top of the annotation provided by NCBI. This software performed a blastp search against the whole downloaded database with default settings and organised the results

into a table with the settings --salltitles -b8 -c1 -p8 --outfmt 6 qseqid sseqid pident evalue stitle.

## eggNOG annotation

The annotated proteome of the assembled genome of *Hydractinia symbiolongicarpus* was downloaded from NCBI[18]. We modified the headers of each gene so it only had their own protein id. The resulting translated genome (from now on referred to as proteome) was queried using EggNOG mapper[86] with the parameters: '-m diamond --sensmode sensitive --target_orthologs all --go_evidence non-electronic' against the EggNOG metazoa database. From the EggNOG output, GO term, functional category COG, and gene name association files, were generated using custom bash code. Full code is available at the project repository.

## Split-seq read processing

Each library was sequenced using a NovaSeq 6000 platform (Illumina) by Novogene (China) and is available for download. The sequencing data comprises the following read counts: 112614850 (9_1), 137678940 (9_2), 139158858 (9_3), 131010556 (9_4), 154135270 (9_5), 578661976 (20_1), 802450388 (20_2), 960633216 (20_3), 669218030 (20_4), 690902500 (27_1), 676659158 (27_2), 807680330 (29_1), and 1026403228 (29_2). Initially, the quality of the output reads was assessed using FastQC (https://www.bioinformatics.babraham.ac.uk/projects/fastqc/). Subsequently, CutAdapt v2.8[87] was employed to remove adaptor sequences, short reads, and low-quality reads. We applied distinct strategies for each pair-end read file. Specifically, the command cutadapt -j 4 -m 60 -q 10 -b AGATCGGAAGAG was executed on each read 1 file to eliminate the Illumina universal adaptor and any reads below 60 bp in length. For read 2 files, we ran the command cutadapt -j 4 -m 94 --trim -n -q 10 -b CTGTCTCTTATA to remove Nextera adaptor sequences, reads shorter than 94 bp, and terminal Ns. Additionally, a "phase" step was performed using the command grep to identify sequences from read 2 files and ensure the corresponding barcodes were in the correct position. Finally, paired reads were generated using pairfq makepairs v0.17 (https://github.com/sestaton/Pairfq) to proceed with further analysis.

The previously assembled reference genome[18] served as the basis for creating a reference database for read mapping. Dropseq_tools-2.3.0 (https://github.com/broadinstitute/Drop-seq/releases/tag/v2.3.0) was then utilised to process the generated GTF file, resulting in the creation of a sequence dictionary, a refFlat file, a reduced GTF file, and corresponding interval files. To build the reference index, we employed STAR-2.7.3a (https://github.com/alexdobin/STAR/releases/tag/2.7.3a) with the parameters --sjdbOverhang 99, --genomeSAindexNbases 13, and --genomeChrBinNbits 14. Each sub-library was individually processed and later combined in the analysis. We leveraged the SPLiT-seq toolbox (https://github.com/RebekkaWegmann/splitseq_toolbox), which incorporates algorithms from Drop-seq_tools-2.3.0, to retrieve, correct, and label the barcodes. The barcodes were labelled with a hamming distance ≤1. For mapping to the reference genome, we used STAR-2.7.3a (https://github.com/alexdobin/STAR/releases/tag/2.7.3a) with --quantMode GeneCounts and all other default settings, except for --outFilterMultimapNmax with 1 and 200 values, which created two modalities of the data and allowed us to retain and analyse reads that mapped up to one and to two hundred different loci in the reference. We employed Picard v2.21.1-SNAPSHOT (https://github.com/broadinstitute/picard) to re-order, merge, align, and tag reads for each sub-library, utilising the SortSam and MergeBamAlignment features. To create expression matrices for each library, we implemented the Drop-seq_tools-2.3.0 features TagReadWithInterval and TagReadWithGeneFunction. We used the feature DigitalExpression from Drop-seq_tools-2.3.0 with the following settings: READ_MQ = 0, EDIT_DISTANCE = 1, MIN_NUM_GENES_PER_CELL = 50, and LOCUS_FUNCTION_LIST = INTRONIC. For further details, the

complete code and documentation can be found in the project repository. The resulting expression matrices, along with the gene models and raw reads, have been uploaded to GEO under the accession code (GSE269914).

## Single-cell analysis

We started processing this dataset with initially 241,340 cells in the "no multimappers" modality and 277,529 in the "with multimappers" modality. We retained both versions of the dataset in the same object following a multimodal integrative strategy with the python package MUON[36]. All the following downstream analyses were performed to the "no multimappers" modality. The processing eliminated genes with high counts using sc.pp.filter_cells with min_counts = 50 and min_genes = 50. We calculated metrics using sc.pp.calculate_qc_metrics, sliced the matrix genes_by_counts <700 and total_counts <750. These steps eliminated 42,227 cells, giving us our final dataset of 199,113 cells. We normalised the matrix using sc.pp.normalize_total with a target_sum = 1e4. We selected high variable genes using sc.pp.highly_variable_genes with n_top_genes = 18000, and sliced the matrix to contain only those genes, storing the raw in an adata.raw object. We then scaled the matrix with sc.pp.scale, performed pca with sc.tl.pca with n_comps = 100, performed a batch correction with the python version of harmony[37] on the 'X_pca' column calculated in the previous step, we constructed a kNN graph with sc.pp.neighbours, with 40 neighbours, 75 principal components, made use of the representation 'X_pca_harmony', and calculated a UMAP visualisation with sc.tl.umap (min_dist = 0.1, spread = 0.5, alpha = 1, gamma = 1.0). We ran the Leiden clustering algorithm using sc.tl.leiden with resolutions 1, 1.5 and 2, which gave 47, 53 and 57 clusters respectively. We calculated marker genes for each cluster using sc.tl.rank_genes_groups, using the clusters obtained with all 3 resolution parameters, and implemented both the Wilcoxon (method = 'wilcoxon') and the Logistic Regression (method = 'logreg'). Following this, we inspected the annotation of marker genes for each cluster and checked for an informative annotation for the top marker genes. We monitored that the size of sets of significant marker genes for each cluster was not smaller than 10 genes, and visually inspecting the cluster sizes. We selected resolution 1.5 for further downstream analyses.

## Compositional analysis

To assess if cell clusters were significantly enriched across the different libraries representing the various colony parts of the animal. We implemented two approaches for testing this, one with the Python package scCODA[88], which uses a Bayesian framework to model cell-type counts while accounting for uncertainty in cell-type proportions and the negative correlative bias via modelling all the joint cell-type proportions. The other approach we also employed was a Fisher exact test. It's important to note that this test is meant for small sample sizes, and it tends to greatly exaggerate *P* values. To account for this statistical effect, we added small positive values, ensuring that variations in enrichment between clusters were not influenced or discarded due to statistical artefacts. For further details, the complete code and documentation can be found in the project repository.

## Count per million (CPM) calculation

A custom Python script (available in the project repository) was used to extract raw counts. This script sliced the raw unprocessed matrix to include only cells present in the processed matrix. Subsequently, cluster information was transferred from the processed matrix to the unprocessed one using a pandas script. To obtain the sum of all counts for each gene within each cluster, numpy was utilised on the matrix. The resulting raw summed counts dataset was then normalised by pseudobulk "library size" using the DESeqDataSetFromMatrix() function, where the parameter design = -condition, and the counts() function with parameter normalised = TRUE from the DESeq2

package[89]. For further details, the complete code and documentation can be found in the project repository.

## Co-occurrence analysis

The cell-type co-occurrence analysis was performed using the tree-FromEnsembleClustering() function, sourced from the provided code[29]. The function was executed with the following parameters: h = c(0.65, 0.95), clustering_algorithm = "hclust", clustering_method = "average", cor_method = "pearson", P = 0.15, n = 1000, bootstrap=FALSE. The analysis performed 1000 iterations of cross-cell-type Pearson correlation, utilising a downsampling of 85% on highly variable genes (FC > 1.5). Subsequently, hierarchical clustering of cell types was performed, and co-occurring pairs of cell types were quantified across iterations, generating a co-occurrence matrix. The final cell-type tree was produced by hierarchically clustering the co-occurrence matrix. For further details, the complete code and documentation can be found in the project repository.

## Transcription factor annotation

The translated proteome of *Hydractinia* was queried for evidence of Transcription Factor (TF) homology using (i) InterProScan[90] against the Pfam[91], PANTHER[92], and (ii) SUPERFAMILY[93] domain databases with standard parameters, (iii) using BLAST reciprocal best hits[94] against swissprot transcription factors[95], and (iv) using OrthoFinder[96] with standard parameters against a set of model organisms (Human, Zebrafish, Mouse, Drosophila) with well-annotated transcription factor databases (following AnimalTFDB v3.0)[97]. For the latter, a given *Hydractinia* gene was counted as TF if at least another TF gene from any of the species belonged to the same orthogroup as the *Hydractinia* gene. The different sources of evidence were pooled together and we kept those *Hydractinia* genes with at least two independent sources of TF evidence. Every TF gene was assigned a class based on its sources of evidence. For complete transparency, the code is available at the project repository.

## Transcription factor analysis

The table of cpms was filtered to retrieve TFs from *Hydractinia*, and gene expression across cell types was scaled and visualised using the ComplexHeatmap R package[98]. To study the expression of each TF class at the cell-type level, we calculated statistics for each TF class. For each class, we looked at how much gene expression varied across different cell types, with the median and average coefficient of variation (CV), the number of genes in that class and the cumulative gene counts. We visualised the relationship between CV and number of genes using the base and ggplot2 packages (https://ggplot2.tidyverse.org) in R v4.1.0 (https://www.R-project.org/).

To represent these findings visually, we quantified the prominence of each TF class in terms of gene counts. For each TF class, we calculated its prominence across different cell clusters by summing up the gene counts in that class within each cluster, and then dividing by the number of genes from that class expressed in that cluster. The resulting matrix was normalised and visualised using a custom ggplot2 wrapper function in R v4.1.0. Fully documented code is available at the project repository.

## WGCNA analysis

We conducted an analysis using Weighted Gene Co-Expression Network Analysis (WGCNA)[50] to elucidate the intricacies of gene interactions in our dataset. We filtered genes which had a coefficient of variation (CV) greater than 1 from the calculated CPM table, and we set the softPower parameter to 14 following the assessment of Scale-Free Topology Model Fit. We then computed adjacency and Topological Overlap Matrices (TOM) using default settings. For the hierarchical clustering of genes, we chose a minimum module size of 75 genes and set the deepSplit parameter to 4. We named and colour-coded all

resulting gene modules manually, in a similar fashion to how cell clusters are designated (Supplementary Data 2 and Supplementary Data 7). These modules were used to reorganise the expression dataset, and the output was visualised using ComplexHeatmap.

To evaluate and represent the association between TF classes and gene modules, we calculated the mean connectivity of each TF gene to the module eigengenes. For each TF class, we calculated the number of genes in that class with a Spearman correlation coefficient equal to or greater than 0.01 with each module eigengene. We normalised the resulting matrix and visualised it using the ComplexHeatmap package. We also filtered the WGCNA results matrix to only show TFs present in the gene modules.

We constructed WGCNA graphs using the TOM matrix, trimming sparse interactions with a low connectedness threshold (>0.01). We employed the igraph package[99] and the Kamada–Kawai layout algorithm[100] for graph visualisation with parameters 'maxiter = 100 * NUM_GENES_GRAPH, kkconst = NUM_GENES_GRAPH', where NUM_GENES_GRAPH is the number of genes present in the analysed graph. We assessed the membership of connected components with the function components() from the igraph package, and their agreement with WGCNA module membership using the adjusted Rand Index implementation adjustedRandIndex() from the package mclust[101].

The analysed graph was divided into subgraphs representing connected components using a custom wrapper function that employs the induced_subgraph() function from the igraph package. Following this, we calculated the centrality of TFs in each subgraph using the closeness() function from the igraph package. Output visualisation was done with ggplot2.

To explore cross-module connections, we created a 'gene × module' matrix that counted how many genes from each module were direct neighbours of a given gene. This matrix was normalised by dividing the number of connections of a gene to each module by the size of the module to which the gene belonged. We aggregated these numbers at the module level to retrieve the number of normalised cross-connections between modules. The resulting matrix was converted into a graph using graph_from_adjacency_matrix() from igraph with parameters 'mode = "upper", weighted = TRUE, diag = FALSE', highlighting the strongest cross-connections based on edge size. Fully documented code is available at the project repository.

## Gene ontology analysis

We conducted Gene Ontology (GO) analyses using the R package topGO v2.52.0[102]. We used the 'elim' method with a custom wrapper function. To keep the analysis stringent, we excluded GO terms with fewer than three genes that were significantly linked to them. Unless stated otherwise, we used all the genes found in *Hydractinia* as the gene universe set to compare against.

## Flanking amino acid analysis of *Shematrin-like* sequences

To assess architecture of the gene set expressed in *Prisilkin-like*+ cells, we developed a custom script searching for glycine repeats (singlets, duplets, triplets quadruplets in $XG_nX$) between flanking amino acids (I, L, V, Y) known for being characteristic of *Shematrins* (see "Code availability")[48,49]. Furthermore, we identified signal peptide regions (N: n-terminal region, H: hydrophobic region, C: c-terminal region) of these genes running a locally installed version SignalP 6.0[69] with default parameters except for 'organism=eukarya, Format=all, mode=slow-sequential'.

## *Stylophora* cell-type gene scores

To understand and compare the evolutionary history of our annotated cell states, we observed clusters from another species which has publicly available single-cell data, *Stylophora pistillata*[29]. We subset lists of genes corresponding to each cell state in *Stylophora* which had at least a value of 1 in their fold change expression. Later, we filtered these genes with

the lists of one-to-one orthologs product of reciprocal blasts between *Hydractinia* and *Stylophora*. The filtered table of gene names in *Hydractinia* was later parsed through our dataset and scored across all cell states with the scanpy function sc.tl.score_genes. Where the distribution of a certain set of genes in each cell group is compared against the same distribution in all other cells not in the group.

## Phylogenetic inference

To assess the conservation of POU family transcription factors we sampled POU domain-containing protein sequences from 14 species, using known human POU protein sequences as Blast queries. Top hits were retained and confirmed by reciprocal Blast. Sequences were trimmed to include only functional domains as identified by NCBI Conserved Domains[103]. Sequences were aligned using Clustal Omega[104] and vacancies and blur sites were removed manually. Phylogenies were inferred by Bayesian analysis using MrBayes[105] under aamodelpr fixed(poisson) with 1 heated chain and 8 cold chains, for 3 million generations.

Following the identification of sequences related to biomineralisation in the atlas of *Hydractinia*, we decided to infer their phylogenetic relationships. For each set of genes, we downloaded the corresponding sequences from Genbank. Identical sequences were removed with CD-HIT[106]. The sequences of each gene were aligned using MAFFT L-INS-i v 7.429[107]. The maximum likelihood topologies were inferred with IQTree 2 v.2.2.0.3[108]. The most appropriate evolutionary model was selected with the Model Finder Plus (-MFP)[109] implementation in IQTree with 1000 replicates of ultrafast bootstrap sampling (-bb)[110]. Visualisation of the final topologies was performed with FigTree v1.4.4 (https://github.com/rambaut/figtree) and IToL v5[111].

## SABER-FISH

SABER-FISH was performed as previously outlined[112], with major modifications to the first two days of the experiment. Raw probe sequences against target mRNAs were designed with OligoMiner and shipped dry from Integrated DNA Technologies. The oligos were pooled after resuspension and simultaneously amplified using hairpin 30 as previously described[113]. Whole feeding polyps were starved for 24 h before amputation, then left overnight. For the stolon in situs, larvae were exposed to seawater containing 20 mM CsCl for metamorphosis on glass slides. After four days, primary polyps showed stolonal extensions. The following day, samples were incubated for 15 min in 4% $MgCl_2$ prior to glass dish fixation, which was done via an initial dropwise addition of 16% paraformaldehyde (PFA) to induce death, followed by incubation of 4% PFA in $MgCl_2$ for 1 h at RT. Tissue samples were then transferred to a 2 mL plastic tube (for the remaining protocol) and washed three times for 5 min each in PBS-0.1% Tween20 (PBSTw), made with diethyl pyrocarbonate-treated water. The samples were then dehydrated in increasing concentrations of ice-cold methanol (MeOH) (in PBSTw) up to 100% MeOH, and permeabilised with increasing concentrations of ice-cold acetone (in MeOH) up to 75% acetone and back down to 100% MeOH for storage overnight. After rehydration, 6 ×10 min washes in ice-cold PBSTw were performed, with a gradual return to RT on the final washes. Following this, a 30 min incubation with 43 °C Whyb buffer (2×SSC pH 7.0, 1% Tween20, 40% Formamide) was performed. This was replaced by prewarmed Hyb1 buffer (2×SSC pH 7.0, 1% Tween20, 40% Formamide, 10% Dextran sulphate) and left for 24 h at 43 °C. The remaining steps of the protocol are outlined in the methods section of Salinas-Saavedra et al.[112]. A 1:1000 dilution of 1 mg/mL DAPI nuclear stain was used instead of Hoechst, for a 45 min incubation. Samples were imaged the day after mounting. All in situ stainings were repeated at least three times with a minimum of 20 animals per replicate.

## Classical ISH

RNA was extracted from mixed adult tissue using Trizol, cleaned using the QIAGEN RNAeasy Mini extraction kit (QIAGEN, 74104) and quantified by Nanodrop, followed by reverse transcription with SuperScript for cDNA synthesis (Invitrogen, 18080051). The desired fragment was amplified using specific primers and cloned into a pGEM-T easy vector (PR-A1360) and confirmed by Sanger sequencing. RNA probe synthesis was performed using SP6 and T7 RNA polymerases according to the manufacturer's protocol (NEB HiScribe T7, E2040S or SP6 High Yield, E2070S, RNA Synthesis Kits). Roche DIG-labelling and detection kit (Roche, 11093657910) was used. In situ hybridisation was performed as previously described[114] except for the following modifications. Proteinase K digestion was replaced with 10 min at 95 °C in PBS. In situ hybridisation was performed at 55 °C. All in situ stainings were repeated at least three times with a minimum of 20 animals per replicate.

## Antibody generation

The anti-Alr1 antibody was generated against recombinant Alr1 protein cytoplasmic tail only. Recombinant protein was produced by cloning the Alr1 cytoplasmic tail (CT) region into a vector, downstream of a His-tagged MBP (maltose-binding protein) and TEV restriction site (Supplementary Data 11) and subsequently expressed by IPTG (Sigma-Aldrich,16758) induction in RIPL cells (Agilent Technologies NC9122855). The resultant recombinant MBP-Alr1CT protein was purified from lysed bacterial cells using Nickel Spin columns (NEB, S1427S). The protein was dialysed into PBS and quantified using Nanodrop. Mice were immunised by intraperitoneal injections with the MBP-Alr1CT protein. Initial immunisation mixture was prepared by mixing equal parts dialysed MBP-Alr1CT protein (at 0.5 mg/mL) and Imject Alum (ThermoFisher, 77161), and incubated at room temperature for 30 min. Each mouse received 250 µL of this mixture. A booster injection was performed one month later with 50 µg protein in 200 µL (1:1 protein and Imject Alum), followed by a second booster one month later. Mice were sacked one month after the final booster. To obtain only recombinant Alr1CT only protein, TEV (NEB, P8112S) digestion was performed overnight followed by removal of MBP and TEV using Nickel Spin columns (NEB, S1427S). Samples containing only Alr1CT protein (flow-through and washes) were concentrated using a 10 K MCWO spin column (ThermoFisher, 88516). The resultant protein was validated at 1:1000 by Western blot (Supplementary Fig. 6), as previously described[115].

## Immunofluorescence staining

For immunofluorescence stainings animals were isolated from colonies by cutting, as previously described, and allowed to regenerate stolons for 3 days. Animals were then anaesthetised in 4% $MgCl_2$ in glass dishes, for 15 min on ice. The 4% $MgCl_2$ was removed leaving only a meniscus over the animals and 60 µL of 16% PFA (Thermo Scientific 28906) in $MgCl_2$ was quickly added dropwise, followed by 2 mL of 4% PFA in $MgCl_2$. Samples were fixed overnight at 4 °C with rocking. Fixative was removed and samples were washed three times with 0.5% Triton in PBS (PBS-T) for 20 min each. Samples were dehydrated by 10 min washes with increasing ethanol concentrations in PBS-T (25%, 50%, 75%, 100%) then stored at −20 °C. For rehydration, the inverse was performed. Samples were permeabilised by three 20 min washes in PBS-T at room temperature, followed by blocking for 1 h in filtered 3% BSA in PBS-T. The Alr1 primary antibody was incubated 1:250 in 3% BSA/PBS-T, overnight at 4 °C with rocking. Samples were then washed three times, for 10 min, in PBS-T at room temperature with rocking. A second block was performed for 15 min in 5% Goat Serum/ 3% BSA in PBS-T at room temperature with rocking. The secondary antibody (Alexa Fluor 594 goat anti-mouse Invitrogen Catalogue # A-11005) was used at a 1:500 dilution in PBS-T and incubated for an hour at room temperature, with rocking. Samples were stained with DAPI (1:1000 dilution of a 1 mg/mL stock) for 1 h and mounted in 2,2′-thiodiethanol (Sigma-Aldrich 166782-100 G)on glass microscope slides, covered with a coverslip and sealed, followed by imaging on a confocal microscope.

All immunofluorescence stainings were repeated at least three times with a minimum of 20 animals per replicate.

## ARC complex analysis
To identify genes in the allorecognition complex (ARC) we carried out protein-protein BLAST searches using the previously identified Alr1 and Alr2 as queries[13]. We confirmed hits by performing reciprocal BLAST. Using the locus of each hit we were able to identify the genomic location which pertained to the ARC.

## Grafting and histology
For identifying allogeneic, incompatible (i.e., Alr-mismatching) colonies, we grafted isolated polyps by stringing them, wound sides facing each other, on a glass capillary. The polyps were forced to contact using two agarose blocks. They were kept at this position for 2 h and were then removed from the capillary and kept in Petri dishes with seawater for up to 24 h. Isogeneic polyps, i.e., polyps from the same colony, were used as controls. For histological analysis, grafts were used 4 h post grafting. Grafts were fixed in 4% formaldehyde, embedded in glycol methacrylate (Sigma-Aldrich 151238), sectioned at 2–5 m, and stained with methylene blue (Sigma-Aldrich M4159). The experiments were repeated at least three times with a minimum of 20 animals per replicate.

## Reporting summary
Further information on research design is available in the Nature Portfolio Reporting Summary linked to this article.

## Data availability
The sc-RNA-seq reads and the cell matrix generated in this study have been deposited in the GEO database under accession code GSE269914 and are also listed in Bioproject PRJNA1124116. The processed annotated dataset was uploaded into the UCSC cell browser [http://cells.ucsc.edu/?ds=hydractinia].

## Code availability
The code used for all the analyses in this study is available in GitHub (https://github.com/scbe-lab/hydractinia_sc_atlas) as well as Zenodo (https://doi.org/10.5281/zenodo.14795685)[116].

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

## Acknowledgements

Research at the Solana lab at Oxford Brookes University and at the Living Systems Institute is supported by MRC grants (MR/S007849/1 and MR/W017539/1), a BBSRC Grant (BB/V014447/1) and a Leverhulme Trust grant (RPG-2019-332 and RPG-2023-330) to J.S. H.G.-C. and E.E. were supported by Nigel Groome studentships from Oxford Brookes University. Work in the Frank lab is funded by a Wellcome Trust Investigator Award in Science (grant no. 210722/Z/18/Z). H.R.H. was a doctoral student at the Science Foundation Ireland Centre for Research Training in Genomic Data Science (grant no. 18/CRT/6214). M.S.-S. was a Human Frontier Science Program Long-Term Postdoctoral Fellow (grant no. LT000756/2020-L). M.E.R. is funded by NERC DTP GW4 +, M.A.-P. is supported by a fellowship from the Fundación General CSIC's Com-Futuro programme funded from the European Union's Horizon 2020 research and innovation programme under the Marie Skłodowska-Curie grant agreement No. 101034263; and M.A.-P. and J.P. are supported by

the Wellcome Trust (210101/Z/18/Z) and the School of Biological Sciences (University of Bristol). Flow cytometry was performed at the Sir William Dunn School of Pathology Flow Cytometry Facility, University of Oxford with the assistance of Dr Robert Hedley and University of Galway Flow Cytometry Core Facility with the assistance of Dr. Shirley Hanley. Technical work related to antibody generation was performed by Dr. Rebecca Elsner, Department of Immunology, University of Pittsburgh. The authors would like to extend thanks to all members of the Solana, Frank and Paps labs, especially Dr Maria Roselló for useful discussion, input and assistance. To Amy Duclaux, Laura Ryan and Cian Lawless for animal care and culturing. Finally, The authors would like to thank Marc Perry and Maximilian Haeussler from the UCSC cell browser for support in uploading the processed dataset and making it publicly available.

## Author contributions

D.A.S.-D, H.R.H., H.G.-C., E.E., N.J.K., M.S.-S., U.F. and J.S. conceived the study and designed the experiments. M.S.-S., H.G.-C., H.R.H., F. and E.E. generated cell dissociations and performed single-cell transcriptomic experiments using *Hydractinia* cell suspensions. D.A.S.-D. and H.R.H. performed bioinformatic analyses, N.J.K. performed preliminary bioinformatic single-cell analyses. A.P.-P. assisted in performing bioinformatic analyses on the transcriptional landscape of *Hydractinia*. R.M.G performed phylogenetic analysis of POU domain proteins. M.E.R. and M.A.-P. supervised by J.P. performed phylogenetic analyses. D.A.S.-D., H.R.H., M.S.-S., F., U.F. and J.S. contributed to the interpretation of the single-cell analysis data, with contributions from all other authors. H.R.H, Y.L.-R., P.K.W., P.H. and C.C. performed cluster validation. M.H.M and S.M.S. generated, validated and provided guidance regarding Alr antibodies. T.F. performed allogeneic grafting and histological analysis. D.A.S.-D., H.R.H., U.F. and J.S. wrote the manuscript and generated the figures, with contributions from all other authors. All authors read and approved the final version of the manuscript.

## Competing interests

The authors declare no competing interests.

## Additional information

[1]Department of Biological and Medical Sciences, Oxford Brookes University, Oxford, UK. [2]Living Systems Institute, University of Exeter, Exeter, UK. [3]Department of Biosciences, University of Exeter, Exeter, UK. [4]Centre for Chromosome Biology, School of Biological and Chemical Sciences, University of Galway, Galway, Ireland. [5]School of Biological Sciences, University of Bristol, Bristol, UK. [6]Institute of Zoology, University of Heidelberg, Heidelberg, Germany. [7]Thomas E. Starzl Transplantation Institute, University of Pittsburgh, Pennsylvania, PA, USA. [8]Department of Biochemistry, University of Otago, Aotearoa, Dunedin, New Zealand. [9]Present address: Stowers Institute for Medical Research, Kansas City, MO, USA. [10]Present address: Department of Systems Medicine, University of Rome Tor Vergata, Rome, Italy. [11]Present address: Institut de Biologia Evolutiva (CSIC-Universitat Pompeu Fabra), Passeig Marítim de la Barceloneta, Barcelona, Spain. [12]Present address: Faculty of Pharmacy, Universitas Muhammadiyah Surakarta, Jawa Tengah, Indonesia. [13]Present address: Sorbonne Université, Institut de Biologie Paris-Seine (IBPS), Paris, France. [14]These authors contributed equally: David A. Salamanca-Díaz, Helen R. Horkan. ✉e-mail: hhorkan@stowers.org; uri.frank@universityofgalway.ie; j.solana@exeter.ac.uk

