## [Peer Review file · Nature Communications]

The Hydractinia cell atlas reveals cellular and molecular principles of cnidarian coloniality

Corresponding Author: Dr Jordi Solana

Version 0:

Reviewer comments:

Reviewer #1

(Remarks to the Author)

This is an exciting, well executed, and well written paper. It provides the most detailed picture yet of cellular composition and expression across zooids and tissues in a colonial organism. This is important because colonial development and organization is so neglected in zoology, but is an exciting opportunity to investigate the origins of new levels of biological organization.

They have a very clear top level finding: "distinct colony parts are characterised primarily by distinct combinations of shared cell types and to a lesser extent by part-specific cell types". Other investigators have looked at histology, bulk RNAseq, and in situ in colonial hydrozoans, but only the single cell methods presented here can answer this basic open question about colony organization and differential expression. We now know that, at least in this species, difference in expression between zooids are driven by different proportions of cells in the zooids, and only to a minimal extent by zooid-specific genes or cell types.

I find the results convincing, and clearly presented. My primary comment is that there are areas to strengthen connections to previous literature, which will make this manuscript even more compelling by placing it in a broader context. In particular, the major findings of the paper are essentially histological - which cells are in which polyps. There is more than a century of work on histology of this animal, so the results should be compared more directly to findings from previous generations of technology. Are the distributions of cell types found in single cell sequencing consistent with what was observed in tissue sections? Here are a couple of papers that are a good jumping off point for that literature:

https://www.researchgate.net/profile/Uri-Frank/publication/225293190_Hydractinia_a_pioneering_model_for_stem_cell_biology_and_reprogramming_somatic_cells_to_pluripotency/links/5625438908aeabddac91c9f7/Hydractinia-a-pioneering-model-for-stem-cell-biology-and-reprogramming-somatic-cells-to-pluripotency.pdf (see references therein)

VAN DE VYVER, G. (1964). Histological studies on development of *Hydractinia echinata*. (Foreign title: Etude histologique du développement d'*Hydractinia echinata* (Flem.) Cahiers de Biologie Marine, Tome V: 295-310.

Even if the cell distribution results using the old methods and single cell sequencing are the same, this paper is still a major advance forward because the expression is itself an additional important story.

(Remarks on code availability)

Reviewer #2

(Remarks to the Author)

The present manuscript from Salamanca-Diaz and co-authors, which presents a cell atlas for the colonial cnidarian *Hydractinia simbiologicarpus*, is a relevant step for cnidarian models and for future studies on cell type evolution. The analyses highlight the cell type composition of the different parts of the colony, showing that the specificity arises from having different proportions of the same cell types. The study characterized transcriptionally well-known cell types (such as i-cell or neurons) and identified two noteworthy novel cell types, the venom-expressing cells (Conodipine+ cells) and biomineralizing cells (Shematrin+ cells). The identification of venom-expressing cells, different from cnidocytes, in the *Hydractinia* colony is interesting, also because an analogous cell type has recently been found in another cnidarian model species, *Nematostella*. The biomineralizing cells open an interesting discussion on the evolution of hard tissues.

The material and methods section is very well described, and the code available on the github repository neatly organised. The analyses are appropriate and insightful. Overall, this study will constitute a useful resource, even if no major breakthrough is provided.

Major comments:

1. The characterization of cell clusters is uniquely in silico. Given that the study identifies some novel cell types - such as the venom-expressing cells - this study would be strengthened by the addition of in situ hybridization data for at least the newly identified cell clusters.
2. The Shematrin-expressing cells are an interesting cell type, and the Authors wisely investigate these genes further. Given that these genes cannot be identified easily by BLAST, a search for "Shematrin" or "Prisilkin" annotation was performed. This strategy is reasonable, but risk skewing the results, as only few cnidarian genomes are publicly annotated. Have the Authors tried any of these strategies, which could strengthen the analyses: i) perform a BLAST search against a dataset of sequenced cnidarian genomes (notably including the colonial hydrozoan *Clytia hemisphaerica*); ii) apply an Orthofinder approach, including molluscan and Brachiopoda species. In case additional analyses were not possible, I recommend toning down the significance.
3. Overall, figures should be clearer. In particular:
 - in figures 3 and 4, the lighter hues of the color-coded names of genes and of cell clusters are difficult to read. I recommend writing all words in black, adding a color-coded dot or bar that recalls the cluster annotation.
 - Figure 5 is difficult to read: lettering is at times very small in size (panels B, G, I), the dots in panel B and G and the legends of panels B, G, I are both difficult to see. The message of panel H is unclear, as it is not evident what these phylogenies are meant to show (e.g. *Hydractinia* genes are not indicated).

Minor points:

4. Please, provide a reference for "All colonial animals exhibit robust regenerative capacities" (line 47).
5. The end of the introduction states that *Hydractinia* has "co-opted a biomineralization gene": are the Authors suggesting that the gene derived from a horizontal gene transfer from a molluscan? Please, clarify this point.
6. How the resolution for the Leiden clustering algorithm was selected (1.5 that gives 53 clusters)? Please explain in the corresponding section of material and methods.
7. The study identifies 53 cell clusters, but both the PAGA and WGCNA analyses identified 38 clusters/co-expression modules. There is therefore a set of clusters that remain "unannotated" (line 495). I recommend thus rephrasing the sentence of discussion "We annotated 53 cell clusters and validated them..." (line 317).
8. Towards the end of the discussion (lines 366-368) a scenario is presented: "Therefore, cell type innovation happens at a low frequency when compared to the innovation of colony parts using preexisting cell types". I do agree with this idea, which I find reasonable, however, I am not convinced that the present study supports this specific conclusion. I recommend re-phrasing or presenting additional evidence from the literature.

(Remarks on code availability)

I could not re-run the code, due to other long analyses already running on my computer - actually it would have been helpful to provide the pre-processed input files. I have inspected the scripts provided, and they are very well organised and clearly annotated. For final publication, I would recommend changing all the "personal" directories and file names (e.g.

"/mnt/sda/david/hydractinia") to generic ones.

Reviewer #3

(Remarks to the Author)

Salamanca-Diaz, Horkan et al. describe single cell types and states in the cnidarian *Hydractinia*, an upcoming model organism that can be used to investigate aspects of coloniality. The study is of significance to the research field of evolutionary developmental biology and even other more applied research fields as well. Below main points of criticisms as well as detailed are summarized:

Main points of criticism:

1. The first main finding, while not entirely surprising ('mix and match'), contrasts with the second key finding of this study—the identification of a cell type potentially involved in biomineralization—which is particularly interesting and warrants further elaboration.

@“mix and match”:

Before accepting the identification of a 'novel' cell type, it is crucial to ask: What truly defines a novel cell type? Such a definition should encompass not just a transcriptomic fingerprint, but also the cell's anatomy, relative position within the organism, and its developmental trajectory—factors that the authors have not fully addressed. Notably, they did not validate the location of these cell types within a zooid using *in situ* hybridization experiments. Similar to how different animal species share numerous cell types due to a common ancestor, it is reasonable to expect that individual zooids in colonial animals might share certain cell types, such as epithelial cells, which are essential for covering the body. However, the proportional abundance of different cell types is a point of interest.

@Biomineralization:

The authors identified 11 biomineralization genes expressed in the shematrin+ cell types, noting their involvement in biomineralization processes in some mollusks and brachiopods. However, at least some of these genes are not direct 1:1 orthologs. For example, while van Willebrand factor A is implicated in mollusk biomineralization, the corresponding gene in *Hydractinia* is VWFD. Similarly, in mollusks, a specific ortholog of chitin synthase (among several) is involved in biomineralization, but it remains unclear which ortholog serves this function in *Hydractinia*. It would have been beneficial if the authors had spent more time properly annotating their phylogenetic trees, including detailed figure legends, and identifying orthologous gene groups. Despite this, the results are still exciting, even if the genes are not exact 1:1 orthologs!

2. The precise location of cell types or states within zooids and stolons remains unknown, as no validation through *in situ* hybridization was performed. It is unclear how the authors determined the location of these cell types. For example, Figure 1E mentions 'Cell type location in the organism,' but it is not clear whether this is based on previously published data or if the authors conducted their own validation and simply omitted it. The authors need to demonstrate that individual zooids or stolons consist of distinct cell types or states and this is only possible by double *in situ* hybridization or HCR (multiplexed). Additionally, it would be beneficial if the authors explained the basis for assuming that a given cell type is, for instance, neuronal. They could reference other published datasets that include *in situ* hybridization results or established cell atlases. The introduction would also benefit from more information about the different types of zooids and stolons. What cell types or states might be expected (as referenced in lines 127 and 129), or have already been described?

3. The results are not described in sufficient detail, making the manuscript more difficult to follow (see specific details below). It would be preferable if the cell types and states of the individual zooids were first presented in separate UMAPs, followed by a merged UMAP (as present). This approach would clarify the distribution of cell types and states right from the beginning.

4. If the zooids are cut from the stolons, it is nearly impossible to make a perfectly clean cut every time, which could result in some tissue contamination between the zooids and stolons. This potential cross-contamination underscores the need for validation *in situ*. Additionally, it would be helpful if the authors provided more information about the size and developmental stage of the animals used in the study. For instance, were younger, still undifferentiated polyps included in the analysis? Clarifying these details would add important context to the results.

Details:

I. 85:
“We studied the genetic profile of that cell 86 type, revealing a cluster of repeat-containing Shematrin-like genes that resemble molluscan shell 87 matrix proteins, raising the possibility that *Hydractinia* has co-opted a biomineralization gene 88 programme to attach to the gastropod shells inhabited by hermit crabs, a key adaptation to their 89 environment.”
This would be a very interesting lead towards a deeper understanding of the role of this putative cell type. Conducting a validation experiment via *in situ* hybridization could provide clear evidence of the exact location of this cell type. Given that stolons are 3D structures, this cell type could potentially be found throughout, perhaps near the substrate where the organism attaches to the gastropod shell, or it might not be present in that region at all.
Results:

I. 247: supp. Fig. 3 deals with pou genes and not shematrins
I. 251 on chromosome 2

Discussion:

I. 308:
please ensure that you refer to cell types and cell states. A progenitor cell would e.g. be a cell state
I. 342: As mentioned above, the authors attempt to demonstrate that colonial animals are composed primarily of shared cell types, with fewer novel ones. However, they do not provide a substantial number of examples to support this claim, and Conodipide cells are one of the few examples for novel cell types they discuss. Rather than delving into a detailed description of these few cell types, the authors write “Further studies are required to fully confirm the identity and function of this broad cell type in *Hydractinia* colonies.” Given these points, I honestly find myself questioning the main goal of this study.

Figures:

Fig. 1 Labels A-F are mixed up and partially missing (F)

Fig. 1a

What is “mixed”? This needs to be labeled in a better way.

Fig.1c

These are single cells of a blend of gastrozooids, gonozooids, the stolon? I don't understand how the color code of these zooids refers to the Umap or is it a Tsne plot? More info is needed.

Fig. 1e: How did the authors define the location of these cell types? Previous papers? Cell types were not validated by *in situ*/ HCR etc.

Also, please define cell type (spermatogenesis) and not the process. What kind of gland cells (there are different types)?

Fig. 1f: Label “F” is missing and it is unclear how the markers of these cell populations refer to the cell types (7 major cell types vs. 8 expression plots)

Fig. 2c

What does 'up in...' mean? Does it indicate that a cell type is more represented in certain zooids? This wording needs to be clarified in the figure legends.

Fig. 5H.

These unrooted phylogenetic trees are uninformative without labeled branches and detailed introductions to the clusters. For example, in the case of carbonic anhydrase, there are different clusters of bilateral sequences, but it's unclear which organisms, gene orthologs, or paralogs they represent. If these trees cannot be thoroughly explained in the main text, then a detailed explanation should be provided in the supplementary information. The relationships depicted in these trees are not immediately obvious.

Fig. 5I “Neurone”? Neurons? Neuron?

Suppl. Material:

Fig. 1F: What is this? Not explained in legend?

Fig. 3

What kind of gene tree? What does it mean “without any modifications”?

Fig. 5

“Gene trees without any modifications of the biomineralization pathway components.”

These are only a few components, and it's known that, for example, in spiralian, only certain chitin synthase genes are involved in biomineralization. Can you distinguish between these different genes? Proper identification is crucial; otherwise, this phylogenetic tree holds little value. Again, these results are exciting but they need to be described more carefully.

Fig. 5h

What are the yellow lines in the protcadherin clade? What are the stippled lines in red?

(Remarks on code availability)

Version 1:

Reviewer comments:

Reviewer #1

(Remarks to the Author)

All of my concerns have been addressed. This is an excellent paper.

(Remarks on code availability)

Reviewer #2

(Remarks to the Author)

In this revised study Salamanca-Diaz, Horkan and co-Authors have addressed carefully all the points raised by the reviewers, either through the manuscript or in the answers to reviewers. I have particularly appreciated the care put in the description of the omics analyses.

Only a fraction of the identified cell types has been validated in situ, however the Authors have chosen the most meaningful ones; they further provided relevant literature supporting the annotation of the known cell types.

The addition of the allerecognition functional assay was unexpected, however it provided an interesting layer of information, and it does strengthen the study. They have additionally generated a custom antibody for Alr1, that will surely be a useful tool for the community.

Minor points:

Please, correct the capital letters in "Ascidians and Cnidarians" (only taxonomical nomenclature should be capitalised, e.g. "Ascidia").

Line 638: "To understand the TFs at a broader level": unclear what this means, please rephrase.

Lines 713-719: The references in this section are indicated as "PMID", please homogenise.

Line 741: please indicate which cDNA was used (e.g. which stage/form) and how it was prepared.

Line 746: how was the recombinant protein produced? Please provide more details.

General comment on Material and Method section: Please indicate catalogue numbers for the reagents in the sections "ACME dissociation", "Flow cytometry", "Classical ISH", "Antibody generation" and "Grafting and histology", where appropriate.

(Remarks on code availability)

Reviewer #3

(Remarks to the Author)

Dear author,

congratulations on this excellent manuscript!

I have just a few comments that should be straightforward to address:

I. 125: "These have known localisation based on morphological description, histology, mRNA

126 in situ hybridisation, immunofluorescence staining and transgenic reporter lines for specific marker genes

127 (Figure 1E, Supplementary Note 1)"

In suppl. Note 1 the authors state "Additionally, all of these clusters co-occur in Supplementary Figure 1D." These are only violine plots that show sth. different.

I. 133. "We validated several of these cell types using in situ hybridisation and

133 immunofluorescence techniques (Supplementary Data 6)."

If downloaded, this file is in Word format and does not include information on validation experiments. Additionally, no data are provided to validate further cell types, not even in the supplementary materials, which appear to contain all available information.

Figure 5 and its legend do not correspond to each other. Double-check also other Figures.

Figure 6H:

Additional information is required. As currently presented in the actual main text body, this tree (or "trees," as mentioned?) does not provide details on the exact sequences used (e.g., GenBank IDs), the species names, or the type of phylogenetic tree. This makes it non-reproducible and limits its usefulness. At the very least, please refer to the supplementary data section where more information might be available.

After all, I would like to emphasize that I consider this manuscript to be of great interest to a broader research community and believe it has the potential to make a valuable contribution to this journal.

(Remarks on code availability)

Open Access This Peer Review File is licensed under a Creative Commons Attribution 4.0 International License, which permits use, sharing, adaptation, distribution and reproduction in any medium or format, as long as you give appropriate credit to the original author(s) and the source, provide a link to the Creative Commons license, and indicate if changes were made. In cases where reviewers are anonymous, credit should be given to 'Anonymous Referee' and the source.

REVIEWER COMMENTS

Reviewer #1 (Remarks to the Author):

This is an exciting, well executed, and well written paper. It provides the most detailed picture yet of cellular composition and expression across zooids and tissues in a colonial organism. This is important because colonial development and organization is so neglected in zoology, but is an exciting opportunity to investigate the origins of new levels of biological organization.

They have a very clear top level finding: "distinct colony parts are characterised primarily by distinct combinations of shared cell types and to a lesser extent by part-specific cell types". Other investigators have looked at histology, bulk RNAseq, and in situs in colonial hydrozoans, but only the single cell methods presented here can answer this basic open question about colony organization and differential expression. We now know that, at least in this species, difference in expression between zooids are driven by different proportions of cells in the zooids, and only to a minimal extent by zooid-specific genes or cell types.

I find the results convincing, and clearly presented. My primary comment is that there are areas to strengthen connections to previous literature, which will make this manuscript even more compelling by placing it in a broader context. In particular, the major findings of the paper are essentially histological - which cells are in which polyps. There is more than a century of work on histology of this animal, so the results should be compared more directly to findings from previous generations of technology. Are the distributions of cell types found in single cell sequencing consistent with what was observed in tissue sections? Here are a couple of papers that are a good jumping off point for that literature:

https://www.researchgate.net/profile/Uri-Frank/publication/225293190_Hydractinia_a_pioneering_model_for_stem_cell_biology_and_reprogramming_somatic_cells_to_pluripotency/links/5625438908aeabddac91c9f7/Hydractinia-a-pioneering-model-for-stem-cell-biology-and-reprogramming-somatic-cells-to-pluripotency.pdf
(see references therein)

VAN DE VYVER, G. (1964). Histological studies on development of *Hydractinia echinata*. (Foreign title: Etude histologique du développement d'*Hydractinia echinata* (Flem.) Cahiers de Biologie Marine, Tome V: 295-310.

Even if the cell distribution results using the old methods and single cell sequencing are the same, this paper is still a major advance forward because the expression is itself an additional important story.

We thank the reviewer for the positive report. We have modified the introduction to include more citations of previous works, added a section to the discussion in which

we provide a historical perspective on i-cell research, and cited other relevant papers throughout the text and Supplementary Note 1.

Reviewer #2 (Remarks to the Author):

The present manuscript from Salamanca-Díaz and co-authors, which presents a cell atlas for the colonial cnidarian *Hydractinia simbiologicarpus*, is a relevant step for cnidarian models and for future studies on cell type evolution. The analyses highlight the cell type composition of the different parts of the colony, showing that the specificity arises from having different proportions of the same cell types. The study characterized transcriptionally well-known cell types (such as i-cell or neurons) and identified two noteworthy novel cell types, the venom-expressing cells (Conodipine+ cells) and biomineralizing cells (Shematrín+ cells). The identification of venom-expressing cells, different from cnidocytes, in the *Hydractinia* colony is interesting, also because an analogous cell type has recently been found in another cnidarian model species, *Nematostella*. The biomineralizing cells open an interesting discussion on the evolution of hard tissues. The material and methods section is very well described, and the code available on the github repository neatly organised. The analyses are appropriate and insightful. Overall, this study will constitute a useful resource, even if no major breakthrough is provided.

Major comments:

1. The characterization of cell clusters is uniquely *in silico*. Given that the study identifies some novel cell types - such as the venom-expressing cells – this study would be strengthened by the addition of *in situ* hybridization data for at least the newly identified cell clusters.

We thank the reviewer for this comment. In light of this, and several other suggestions, we sought to validate some of the cell types we identified in the study. Specifically we provide validation for two of the Conodipine+ cell types, one newly renamed 'Avidin+' cells (cluster 13, previously 'Cdpi3+' cells) validated by SABER-FISH *in situ* (Figure 1G), and one newly renamed 'Alr1+' cells (cluster 9, previously 'Cdpi2+' cells) validated by IF against Alr1 protein (Figure 5C and 5D). We also validated cluster 24, 'gl Chitinase2+' cells by colourimetric *in situ* against Chitinase2 mRNA. Finally we identified the stolon enriched cell type (previously named 'Shematrín+', now named 'Prisilkin+'), using SABER-FISH with probes against Prisilkin mRNA. We find the label concentrated in the stolon at the attachment point of the base of the polyp (Figure 6I).

2. The Shematrín-expressing cells are an interesting cell type, and the Authors wisely investigate these genes further. Given that these genes cannot be identified easily by BLAST, a search for "Shematrín" or "Prisilkin" annotation was performed. This strategy is reasonable, but risk skewing the results, as only few cnidarian genomes are publicly annotated. Have the Authors tried any of these strategies, which could strengthen the analyses: i) perform a BLAST search against a dataset of sequenced cnidarian genomes (notably including the colonial hydrozoan *Clytia*

hemisphaerica); ii) apply an Orthofinder approach, including molluscan and Brachiopoda species. In case additional analyses were not possible, I recommend toning down the significance.

We thank the reviewer for these suggestions. We did a blast search against several annotated and unannotated cnidarian genomes, including *Clytia*. Unfortunately this did not result in any significant match. Moreover, we implemented an Orthofinder approach including several species within the context of our transcription factor annotation. In this analysis we included cnidarians, molluscs, annelids, brachiopods, platyhelminthes, arthropods, chordates, echinoderms, placozoans, poriferans, ctenophores, and opisthokonts. In the results we found that *Shematrins* and *Prisilkins* indeed are in orthogroups but only with either uncharacterized genes from *Hydractinia echinata* or several cnidarian uncharacterised genes. Thus, these orthogroups do not include other biomineralizing phyla (e.g. molluscs or brachiopoda). In the end, we resorted to reporting genes annotated as *Prisilkin* or *Shematrins* in NCBI annotated genomes as a standardised analysis. Further analyses are needed to elucidate the evolutionary history of *Prisilkin* and *Shematrins* genes.

3. Overall, figures should be clearer. In particular:

- in figures 3 and 4, the lighter hues of the color-coded names of genes and of cell clusters are difficult to read. I recommend writing all words in black, adding a color-coded dot or bar that recalls the cluster annotation

We thank the reviewer for this suggestion. This has been modified in the figures and in each corresponding legend so is readable for the reader and with a clearer message.

- Figure 5 is difficult to read: lettering is at times very small in size (panels B, G, I), the dots in panel B and G and the legends of panels B, G, I are both difficult to see. The message of panel H is unclear, as it is not evident what these phylogenies are meant to show (e.g. *Hydractinia* genes are not indicated).

We agree that the previous figure was difficult to read. To clarify our message we have modified the figure to make panel H contain only one phylogeny and removed the rest to the supplement (Supplementary Data 11). We have clarified the annotation of this phylogeny and modified the legend accordingly. By removing the additional phylogenies we were able to increase the size of the remaining panels which makes them more easily readable.

Minor points:

4. Please, provide a reference for “All colonial animals exhibit robust regenerative capacities” (line 47).

The text has been modified to more accurately detail those phyla for which we have evidence of regenerative capacity in colonial animals; we have also included additional references to support this point.

This section now reads:

“Many colonial animals, such as Ascidians and Cnidarians, exhibit robust regenerative capacities that are based on populations of adult stem/stem-like cells 5, 9, 10.”

5. The end of the introduction states that Hydractinia has “co-opted a biomineralization gene”: are the Authors suggesting that the gene derived from a horizontal gene transfer from a molluscan? Please, clarify this point.

We did not mean to imply that the gene arose in Hydractinia due to a horizontal gene transfer event, rather that a gene which is used for one function in a species, is used for a different function in Hydractinia. We use the word co-opted here to highlight the different but adjacent roles of the gene between the organisms.

6. How the resolution for the Leiden clustering algorithm was selected (1.5 that gives 53 clusters)? Please explain in the corresponding section of material and methods.

We thank the reviewer for noting this point. Indeed the number of clusters that are defined in any single-cell analysis to a certain extent arbitrary, and so is the cluster resolution used for all the analysis and interpretations. For this reason, we tried to make an informed decision on the clustering based primarily on the annotations of the marker genes in each resolution. Initially, we considered resolutions 1 to 6, and after visually examining the clustering, we narrowed it down to 1, 1.5 and 2 (resolutions shown here, which can also be found in the github repository). We carefully considered the markers obtained, the potential clusters that could be products of overclustering or underclustering, as well as the ones that could not be annotated. We performed this over a number of iterations of the analysis before generating the analyses performed here. In the end, we chose leiden 1.5 as this resolution capitulates a number of previously identified cell types in the animal, including *RFamide* and *GLWamide* neurones, two types of nematocytes, the separation of i-cells and early progenitors and does not cluster epithelial and epitheliomuscular cells into subclusters devoid of unique and identifiable marker genes. Additionally this resolution resolved other clusters that were subject of analysis (e.g. Conodipine+ cells, Prsilkin-like+ cells among others). This has been modified and explained in the materials and methods section.

7. The study identifies 53 cell clusters, but both the PAGA and WGCNA analyses identified 38 clusters/co-expression modules. There is therefore a set of clusters that remain “unannotated” (line 495). I recommend thus rephrasing the sentence of discussion “We annotated 53 cell clusters and validated them...” (line 317).

We have modified the text to clarify that 53 clusters were identified and 38 were annotated as specific cell types while the rest remain unannotated, and that the 38 with cell type annotations were further validated.

The section now reads: “We identified 53 cell clusters, 38 of which we annotated as specific cell types, and 15 of which remain unannotated. We validated the 38 cell type annotations through several *in silico* methodologies.”

8. Towards the end of the discussion (lines 366-368) a scenario is presented: “Therefore, cell type innovation happens at a low frequency when compared to the innovation of colony parts using preexisting cell types”. I do agree with this idea, which I find reasonable, however, I am not convinced that the present study supports this specific conclusion. I recommend re-phrasing or presenting additional evidence from the literature

We apologise for the confusion. We presented this as a hypothesis (“We propose that cell type innovation...”), and we believe that our study supports this proposal, given that most cell types in the specific body parts are shared (“mix and match”) and part-specific types are rarer. As it is a hypothesis, further studies are needed to test it. To strengthen this, we have added a sentence.

The section now reads: “Therefore, we propose that cell type innovation happens at a low frequency when compared to the innovation of colony parts using preexisting cell types. Future single cell studies in closely related species will test this hypothesis. Nonetheless, cell type innovation might have played important roles in the evolution of coloniality in cnidarians. In the future, single-cell techniques will enable the

identification of additional biomineralization cell types and other cell type innovations, allowing further investigation into the cellular and molecular basis of adaptation”

Reviewer #2 (Remarks on code availability):

I could not re-run the code, due to other long analyses already running on my computer - actually it would have been helpful to provide the pre-processed input files. I have inspected the scripts provided, and they are very well organised and clearly annotated. For final publication, I would recommend changing all the "personal" directories and file names (e.g. `"/mnt/sda/david/hydractinia/"`) to generic ones.

We thank the reviewer for this suggestion. We uploaded the processed dataset into the UCSC cell browser, so it can be accessed by the whole community and the website has been included in the data availability section of the manuscript. Together with this, we modified the paths on the code repository to '\$PATH'.

Reviewer #3 (Remarks to the Author):

Salamanca-Díaz, Horkan et al. describe single cell types and states in the cnidarian *Hydractinia*, an upcoming model organism that can be used to investigate aspects of coloniality. The study is of significance to the research field of evolutionary developmental biology and even other more applied research fields as well. Below main points of criticisms as well as detailed are summarized:

Main points of criticism:

1. The first main finding, while not entirely surprising ('mix and match'), contrasts with the second key finding of this study—the identification of a cell type potentially involved in biomineralization—which is particularly interesting and warrants further elaboration.

@“mix and match”:

Before accepting the identification of a 'novel' cell type, it is crucial to ask: What truly defines a novel cell type? Such a definition should encompass not just a transcriptomic fingerprint, but also the cell's anatomy, relative position within the organism, and its developmental trajectory—factors that the authors have not fully addressed.

The subject of cell type evolution, encompassing questions such as what is a cell type, what is homology of cell types, and what is therefore a truly novel cell type is a matter of very active discussion in the literature today, with conflicting opinions. For instance, see Xia and Yanai 2019 (PMID: 31249003) or Domcke and Shendure 2023 (PMID: 36931241). In this manuscript, rather than adhering to a definition of cell type, which

could allow us to extrapolate a definition of “novel cell type”, we are limiting to a “transcriptomic similarity” approach. As a result of this, we only suggest the scenario of cell type innovation being rarer than the “mix and match” of preexisting cell types. We have rephrased some instances to clarify that these are suggestions.

Notably, they did not validate the location of these cell types within a zooid using in situ hybridization experiments. Similar to how different animal species share numerous cell types due to a common ancestor, it is reasonable to expect that individual zooids in colonial animals might share certain cell types, such as epithelial cells, which are essential for covering the body. However, the proportional abundance of different cell types is a point of interest.

According to this and other suggestions we sought to validate several clusters, with specific focus on the cell types that were newly identified in this study. Specifically we provide validation for two of the Conodipine+ cell types, one newly renamed ‘Avidin+’ cells (cluster 13, previously ‘Cdpi3+’ cells) validated by SABER-FISH in situ (Figure 1G) showing localised expression at the base of feeding polyp tentacles. One newly renamed ‘Alr1+’ cells (cluster 9, previously ‘Cdpi2+’ cells) validated by IF against Alr1 protein (Figure 5C and 5D) showing expression only in the epidermis, and with a higher abundance of these cells in the distal end of the stolon. We also validated cluster 24, ‘gl Chitinase2+’ cells by colourimetric in situ against Chitinase2 mRNA, highlighting the presence of these cells in the gastrodermis of the feeding polyp and distributed throughout the stolon. Finally we identified the stolon enriched cell type (previously named ‘Shematrinn+’, now named ‘Prisilkin+’), using SABER-FISH with probes against Prisilkin mRNA. We find the label concentrated in the stolon at the attachment point of the base of the polyp (Figure 6I). Here we highlight a range of cell types that are variably distributed across the colony compartments.

@Biomineralization:

The authors identified 11 biomineralization genes expressed in the shematrinn+ cell types, noting their involvement in biomineralization processes in some mollusks and brachiopods. However, at least some of these genes are not direct 1:1 orthologs. For example, while van Willebrand factor A is implicated in mollusk biomineralization, the corresponding gene in *Hydractinia* is VWFD. Similarly, in mollusks, a specific ortholog of chitin synthase (among several) is involved in biomineralization, but it remains unclear which ortholog serves this function in *Hydractinia*. It would have been beneficial if the authors had spent more time properly annotating their phylogenetic trees, including detailed figure legends, and identifying orthologous gene groups. Despite this, the results are still exciting, even if the genes are not exact 1:1 orthologs!

We thank the reviewer for this comment, we have sought to greatly improve the clarity of our phylogenetic tree annotation and corresponding figure legends. Additionally, we have removed 3 of the trees to the supplement, to bring focus on the a-carbonic-anhydrase phylogeny, allowing us to more clearly display this relationship. We have

also modified the text to more clearly convey the steps taken to arrive at the analysis of these genes. In short, our interest in these genes involved in biomineralization arose when they were returned as a result in our WGCNA analysis, rather than from taking a candidate gene approach. Due to this, not all genes that may be involved in biomineralization will be present here, especially as this was a focus on a single cell type, the 'prisilkin+' cells, and other cell types may (and likely are) involved in this process, expressing their own complements of genes. Finally, we agree with the reviewer that it would be interesting to assess the function of these genes in *Hydractinia*, but functional studies are beyond the scope of this study, not least due to the technical difficulty of working with stolon samples.

2. The precise location of cell types or states within zooids and stolons remains unknown, as no validation through *in situ* hybridization was performed. It is unclear how the authors determined the location of these cell types. For example, Figure 1E mentions 'Cell type location in the organism,' but it is not clear whether this is based on previously published data or if the authors conducted their own validation and simply omitted it.

We apologise for this misunderstanding. We had described the origin of our samples in the text associated to Figure 1, but then analysed the proportions in Figure 2 and associated text. As a result, that section was unclear. Our single cell samples contain isolated stolons, sexual polyps and feeding polyps. Thus, one can compare the proportions obtained in each of these samples. This is where our location statements come from. The locations mentioned in Figure 1E are indeed extracted from the literature. We have added those citations to the main text. Furthermore, as explained below, we have now validated several cell types using *in situ* techniques.

To make this clearer, the opening sentence of the section related to Figure 2 now reads:

" We then aimed to ascertain the cellular composition of each individual colony part. We examined the contribution of each sample, including manually isolated stolons as well as sexual and feeding polyps, to our integrated analysis, revealing key differences in their cellular composition"

The authors need to demonstrate that individual zooids or stolons consist of distinct cell types or states and this is only possible by double *in situ* hybridization or HCR (multiplexed). Additionally, it would be beneficial if the authors explained the basis for assuming that a given cell type is, for instance, neuronal. They could reference other published datasets that include *in situ* hybridization results or established cell atlases. The introduction would also benefit from more information about the different types of zooids and stolons. What cell types or states might be expected (as referenced in lines 127 and 129), or have already been described?

As stated on the corresponding section (“Cellular composition of Hydractinia colony parts”) cell types are enriched in the different compartments rather than unique to any one compartment. Thanks to useful comments raised by the referees we have validated several cell types in the manuscript, one of which is the stolon enriched type, Prsilkin+ cells (previously named Shematrין+ cells). Our in situ validation shows the cells are located in the stolon around the base of the polyp at the attachment point to the substrate, and not in the feeding polyp itself, supporting our statistical analysis which shows this cell type is highly enriched in the stolon. Another of the highly enriched cell types, ‘gametogenesis cells’, marked by histone H2b3 and 4, is shown to be highly enriched in the sexual polyp. This finding is in line with the literature which shows by in situ of Histone H2b3/4 that spermatogenesis is, unsurprisingly, enriched in the sexual polyp. Our rationale for the naming of each of the clusters can be found in the Supplementary Note and here we detail the specific marker genes identified and cite all relevant literature that helped us arrive at our cell type classifications. Much of this literature includes in situ, IF or transgenic reporter validation of these cell types.

3. The results are not described in sufficient detail, making the manuscript more difficult to follow (see specific details below). It would be preferable if the cell types and states of the individual zooids were first presented in separate UMAPs, followed by a merged UMAP (as present). This approach would clarify the distribution of cell types and states right from the beginning.

We would like to thank the reviewer for this suggestion. We have added individual analyses for zooid types (feeding and sexual polyps) as well as stolons in our new Supplementary Figure 2. These give, as the referee suggests, an idea of the representation of the individual samples. Unfortunately, UMAPS are just a way of representing the highly dimensional data in 2D, but have shortcomings when it comes to a more quantitative analysis. The density of the dots is rarely homogeneous, and as a result, the UMAP can give a false impression of the relative abundances. Furthermore, and as the reviewer says later, our samples are not 100% pure, and stolon samples and polyp samples will be contaminated with cells coming from other parts. Therefore, the individual parts contain all cell clusters, albeit at different proportions. As a result, in order to ascertain the cellular composition of each colony part we need to perform a statistical analysis of the cell type ratios, which is what we do in Figure 2 and its associated text.

Presenting the separated clusters of stolons and the two polyp types before the merged dataset is complicated because of several reasons: a) Our merged dataset contains “mixed polyps”, a sample that adds numbers to our dataset and aids in cluster identification. b) The clustering of the merged dataset has to be done prior to the clustering of the types, so that the clusters are comparable. The three separated clusterings have different numbers of clusters, and while most are one-to-one equivalent, many others will represent one cluster in one colony part and two in another colony part, complicating the analysis.

As a result of this, the revised version keeps Figure 1 with the merged dataset, but then introduces the colony parts as individual analyses with their own clusters in Supplementary Figure 2, to then statistically analyse the proportions in Figure 2.

4. If the zooids are cut from the stolons, it is nearly impossible to make a perfectly clean cut every time, which could result in some tissue contamination between the zooids and stolons. This potential cross-contamination underscores the need for validation via in situ. Additionally, it would be helpful if the authors provided more information about the size and developmental stage of the animals used in the study. For instance, were younger, still undifferentiated polyps included in the analysis? Clarifying these details would add important context to the results.

We agree with the reviewer that the interface between polyp and stolon is not 'clean cut', hence our mindful decision to refer to cell types as enriched between colony parts, rather than specific to colony parts. This is a two-fold logic, the first being that we cannot determine morphologically where the polyp ends and the stolon begins, and second that a harsh delineation likely does not exist in the first place. Regarding the comment on validation, see previous responses detailing our efforts to validate novel cell types. We agree that the in situ validations have helped with this, especially in the case of the Prsilkin+ cells which are located in the stolon at the base of the polyp, but absent from the polyp itself, showing that these cells are not unlikely to fall victim to inclusion in the polyp dataset if cuts were made too 'low' on the polyp. Regarding the 'age' of the animals, this study exclusively used male clone 291-10 which is an established lab clone in excess of 10 years old at least (based on date of collection, in 2014). Polyp samples were isolated from sexually mature colonies and we aimed to sample 'large' polyps (newly budded polyps tend to be smaller in size) - we have modified the methods to reflect this.

Details:

I. 85: "We studied the genetic profile of that cell
86 type, revealing a cluster of repeat-containing Shematin-like genes that resemble molluscan shell
87 matrix proteins, raising the possibility that Hydractinia has co-opted a biomineralization gene
88 programme to attach to the gastropod shells inhabited by hermit crabs, a key adaptation to their
89 environment."

This would be a very interesting lead towards a deeper understanding of the role of this putative cell type. Conducting a validation experiment via in situ hybridization could provide clear evidence of the exact location of this cell type. Given that stolons are 3D structures, this cell type could potentially be found throughout, perhaps near the substrate where the organism attaches to the gastropod shell, or it might not be present in that region at all.

We thank the reviewer for this insightful comment, thanks to this comment and others we sought to validate the presence and distribution of a number of the novel cell types we identified in this study, including the 'Prsilkin+' cells (previously 'Shematin+' cells). We validated this cell type by SABER-FISH with probes against Prsilkin mRNA

(Figure 6I). The reviewer was correct in that this revealed further insights into the biology of this stolon-enriched, novel, cell type. We saw the probe signal focused at the point of attachment of the polyp to the substrate - visualized by imaging both the feeding polyp and stolon from an aboral view (aka from underneath the stolon). We now show in the manuscript that this cell type is stolon enriched (not present in the polyp), specifically at the point of attachment, shedding light on the role of this cell type and supporting our hypothesis that these cells play a role in interfacing the colony with the gastropod shells which they inhabit.

Results:

I. 247: supp. Fig. 3 deals with pou genes and not shematrixins

Corrected

I. 251 on chromosome 2

Corrected

Discussion:

I. 308:

please ensure that you refer to cell types and cell states. A progenitor cell would e.g. be a cell state

We have clarified when we refer to cell types vs cell states, however the delineation between a cell type and a cell state becomes increasingly complex, especially in organisms with pluripotent stem cells that continually replenish all cell types in the adult organism, with the exception of terminally differentiated cell types, all cells in the animal are in varying 'states' of differentiation.

I. 342: As mentioned above, the authors attempt to demonstrate that colonial animals are composed primarily of shared cell types, with fewer novel ones. However, they do not provide a substantial number of examples to support this claim, and Conodipide cells are one of the few examples for novel cell types they discuss. Rather than delving into a detailed description of these few cell types, the authors write "Further studies are required to fully confirm the identity and function of this broad cell type in Hydractinia colonies." Given these points, I honestly find myself questioning the main goal of this study.

We thank the reviewer for this comment. Our revised manuscript further delves into the description of these cell types, providing localisation details using image-based approaches, deepening in the genomic and transcriptomic characterisation, and providing functional experiments. Specifically:

- The *Conodipin+* *Alr1+* cells are located in epithelial tissue, expressing many *Alr* related genes. We support their function as mediators of the allorecognition response by performing grafting experiments that show that epithelial tissue, and not gastrodermal tissue, exerts this response.

- The *Prisilkin+* cells are located in stolons, accumulating at the base of the polyps, and have a biomineralization gene profile, including secreted peptides that resemble molluscan shell proteins. This suggests that they generate shell-like material to add the colony in their adhesion to the molluscan shell substrate. Furthermore we show that this cell type is not transcriptionally similar to coral calcicoblasts and is therefore likely an evolutionary morphology of the *Hydractinia* clade.

We believe that these findings provide key details about the cellular and molecular basis of these two functions (Allorecognition and Stolon adhesion to the substrate) that are fundamental for coloniality. However, we find that it is still true that further studies are required to delve deeper into the biology of these cell types, and the context of their evolutionary origin, studies that will be enabled by our *Hydractinia* Cell Atlas project.

Figures:

Fig. 1 Labels A-F are mixed up and partially missing (F)

Corrected

Fig. 1a

What is “mixed”? This needs to be labeled in a better way.

We have modified the legend for Figure 1A accordingly.

Fig.1c

These are single cells of a blend of gastrozooids, gonozooids, the stolon? I don't understand how the color code of these zooids refers to the Umap or is it a Tsne plot? More info is needed.

We have removed the tissue labelling image as we agree it added confusion, the UMAP shows all of the sublibraries together and is coloured by cluster/annotated broad cell type.

Fig. 1e: How did the authors define the location of these cell types? Previous papers? Cell types were not validated by in situs/ HCR etc.

The location of the cell types shown in Figure 1E was indeed based on previous literature, the citations for which can be found in the cluster annotation logic outlined in Supplementary Note 1. We have modified the text to more explicitly detail the

methodologies which had previously been used to identify these cell types and added a tag to Supplementary Note 1 to guide the reader to it.

Also, please define cell type (spermatogenesis) and not the process. What kind of gland cells (there are different types)?

We have renamed this cluster 'gametogenesis cells'. This cluster includes cells in various states of the gametogenesis process which share many global markers but could also subcluster into other cell types - based on the Hydractinia atlas in Schnitzler et al 2024 [PMID: 37786714]. Proportionally our dataset includes fewer cells from the spermatogenesis lineage than does the Schnitzler et al dataset. As such Schnitzler et al were able to more closely refine their analysis of this trajectory. In order to avoid misnaming this broader cluster in our analyses we refer to them as a grouped cell type - spermatogenesis cells, which likely includes cells at varying stages from germ to sperm.

Regarding gland cells, we have modified the supplementary note to include the naming of these clusters under the common broad type 'gland'. Within this we highlight the two types, mucosal and digestive gland cells. Gland cells are an understudied cell type in Hydractinia and as we do not further analyse them in this manuscript, we identified these differences based only on our computational analyses, so we chose to refer to them only at the broader level of gland cells.

Fig. 1f: Label "F" is missing and it is unclear how the markers of these cell populations refer to the cell types (7 major cell types vs. 8 expression plots)

We have modified the legend to correct the error and increase clarity.

Fig. 2c

What does 'up in...' mean? Does it indicate that a cell type is more represented in certain zooids? This wording needs to be clarified in the figure legends.

This has been modified in each corresponding legend of Figure 2C, D and E. The expression 'up in...' has been modified in the legend explaining the credible enrichment in each of the comparisons and making a better explanation for the reader.

Fig. 5H.

These unrooted phylogenetic trees are uninformative without labeled branches and detailed introductions to the clusters. For example, in the case of carbonic anhydrase, there are different clusters of bilaterian sequences, but it's unclear which organisms, gene orthologs, or paralogs they represent. If these trees cannot be thoroughly explained in the main text, then a detailed

explanation should be provided in the supplementary information. The relationships depicted in these trees are not immediately obvious.

Thank you for this comment. We agree and have significantly modified the figures including phylogenetic trees. Firstly we have moved 3 out of 4 of the trees to the Supplementary Data 11. This has allowed us to increase the size of the α -carbonic-anhydrase tree in the main figure and to focus our message. We have added additional annotations in the main figure and included annotations of all species in the supplementary figures of each of the trees (Supplementary Data X). We have also significantly increased the figure legends to add clarity.

Fig. 5I “Neurone”? Neurons? Neuron?

Neurone is an accepted British English spelling - we have been consistent with our use of British English throughout the manuscript.

Suppl. Material:

Fig. 1F: What is this? Not explained in legend?

The heatmap in Supplementary Figure 1F refers to the co-occurrence analysis explained in the methods section and helped us in the final cluster annotation and establish relationships between cell types. This has been modified in the legend of the figure.

Fig. 3

What kind of gene tree? What does it mean “without any modifications”?

Legend has been updated to clarify this is a Bayesian tree with full length POU sequences - not only POU domains.

Fig. 5

“Gene trees without any modifications of the biomineralization pathway components.”

These are only a few components, and it’s known that, for example, in spiralian, only certain chitin synthase genes are involved in biomineralization. Can you distinguish between these different genes? Proper identification is crucial; otherwise, this phylogenetic tree holds little value. Again, these results are exciting but they need to be described more carefully.

We thank the reviewer for raising this concern. Indeed phylogenetic trees without proper description makes the reader get lost in the results. In the manuscript we sought to rectify that statement indicating these genes may be involved in the biomineralization pathway of *Hydractinia* based on a phylogenetic tree annotation of components with high incidence of appearing in the biomineralization pathway. We have modified this figure focusing only in α -carbonic anhydrase, and deepening

the description of the methods leading to these results in the figure legends and the corresponding supplementaries.

Fig. 5h

What are the yellow lines in the protcadherin clade? What are the stippled lines in red?

The figure has been modified.

Reviewer #1 (Remarks to the Author):

All of my concerns have been addressed. This is an excellent paper.

We would like to thank reviewer 1 for their positive comments and contribution to improving our manuscript.

Reviewer #2 (Remarks to the Author):

In this revised study Salamanca-Diaz, Horkan and co-Authors have addressed carefully all the points raised by the reviewers, either through the manuscript or in the answers to reviewers. I have particularly appreciated the care put in the description of the omics analyses.

Only a fraction of the identified cell types has been validated in situ, however the Authors have chosen the most meaningful ones; they further provided relevant literature supporting the annotation of the known cell types.

The addition of the allorecognition functional assay was unexpected, however it provided an interesting layer of information, and it does strengthen the study. They have additionally generated a custom antibody for Alr1, that will surely be a useful tool for the community.

We appreciate Reviewer 2's comments for their positive feedback and valuable contributions to enhancing our manuscript. We have addressed their minor points in the manuscript, as explained in the comments below.

Minor points:

Please, correct the capital letters in "Ascidians and Cnidarians" (only taxonomical nomenclature should be capitalised, e.g. "Ascidia").

We have changed this

Line 638: "To understand the TFs at a broader level": unclear what this means, please rephrase.

We have modified this sentence to further explain it. It now reads "*To study the expression of each TF class at the cell type level, we calculated statistics for each TF class*"

Lines 713-719: The references in this section are indicated as "PMID", please homogenise.

We apologise for this mistake; these references were left unformatted in the submitted version. This has been corrected now.

Line 741: please indicate which cDNA was used (e.g. which stage/form) and how it was prepared.

We have added further details into this section. The section now reads:

“RNA was extracted from mixed adult tissue using Trizol, cleaned using the QIAGEN RNAsasy Mini extraction kit (QIAGEN, 74104) and quantified by Nanodrop, followed by reverse transcription with SuperScript for cDNA synthesis (Invitrogen, 18080051). The desired fragment was amplified using specific primers and cloned into a pGEM-T easy vector (PR-A1360) and confirmed by Sanger sequencing. RNA probe synthesis was performed using SP6 and T7 RNA polymerases according to the manufacturer’s protocol (NEB HiScribe T7, E2040S or SP6 High Yield, E2070S, RNA Synthesis Kits). Roche DIG-labelling and detection kit (Roche, 11093657910) was used. In situ hybridization was performed as previously described¹¹⁴ except for the following modifications. Proteinase K digestion was replaced with 10 min at 95°C in PBS. In situ hybridization was performed at 55°C.”

Line 746: how was the recombinant protein produced? Please provide more details.

We apologise for this omission. We have provided more details as requested. We also included the sequence of the recombinant protein expressing vector in Supplementary Data (11). The section now reads:

“The anti-Alr1 antibody was generated against recombinant Alr1 protein cytoplasmic tail only. Recombinant protein was produced by cloning the Alr1 cytoplasmic tail (CT) region into a vector, downstream of a His-tagged MBP (maltose-binding protein) and TEV restriction site (Supplementary data 11) and subsequently expressed by IPTG (Sigma-Aldrich,16758) induction in RIPL cells (Agilent Technologies NC9122855). The resultant recombinant MBP-Alr1CT protein was purified from lysed bacterial cells using Nickel Spin columns (NEB, S1427S). The protein was dialyzed into PBS and quantified using Nanodrop. Mice were immunized by intraperitoneal injections with the MBP-Alr1CT protein. Initial immunisation mixture was prepared by mixing equal parts dialyzed MBP-Alr1CT protein (at .5mg/mL) and Imject Alum (ThermoFisher, 77161), and incubated at room temperature for 30 minutes. Each mouse received 250uL of this mixture. A booster injection was performed one month later with 50 ug protein in 200 uL (1:1 protein and Imject Alum), followed by a second booster one month later. Mice were sacked one month after the final booster. To obtain only recombinant Alr1CT only protein, TEV (NEB, P8112S) digestion was performed overnight followed by removal of MBP and TEV using Nickel Spin columns (NEB, S1427S). Samples containing only Alr1CT protein (flow-through and washes) were concentrated using a 10K MCWO spin column (ThermoFisher, 88516). The resultant protein was

validated at 1:1000 by Western blot (Supplementary Figure 6) as previously described [115]

General comment on Material and Method section: Please indicate catalogue numbers for the reagents in the sections “ACME dissociation”, “Flow cytometry”, “Classical ISH”, “Antibody generation” and “Grafting and histology”, where appropriate.

We have added the requested information where appropriate.

Reviewer #3 (Remarks to the Author):

Dear author,

congratulations on this excellent manuscript!

I have just a few comments that should be straightforward to address:

We appreciate Reviewer 3's thoughtful feedback and constructive suggestions. We have incorporated their minor points into the manuscript, as detailed in our responses below.

I. 125: “These have known localisation based on morphological description, histology, mRNA

126 in situ hybridisation, immunofluorescence staining and transgenic reporter lines for specific marker genes

127 (Figure 1E, Supplementary Note 1)”

In suppl. Note 1 the authors state “Additionally, all of these clusters co-occur in Supplementary Figure 1D.” These are only violine plots that show sth. different.

We apologise for this mistake. The co-occurrence analysis is in Supplementary Figure 1F, not 1D. We have corrected this in both instances.

I. 133. “We validated several of these cell types using in situ hybridisation and 133 immunofluorescence techniques (Supplementary Data 6).”

If downloaded, this file is in Word format and does not include information on validation experiments. Additionally, no data are provided to validate further cell types, not even in the supplementary materials, which appear to contain all available information.

We have rephrased this to enhance the clarity. The word file only applies to the probes and its design based on marker genes (and it does not apply to immunofluorescence), so we have made this more explicit by editing the place of the supplementary data calling.

The section now reads:

"We validated several of these cell types with in situ hybridisation using probes designed to target their marker genes (Supplementary Data 6) as well as immunofluorescence techniques"

As per the validation of other cell types, this can be done by designing probes against more markers – we publish the entire list of markers by two methods as well as the WGCNA module genes. The IDs in all these lists correspond to the publicly available *Hydractinia* genome (NCBI).

Figure 5 and its legend do not correspond to each other. Double-check also other Figures.

We apologise for this mistake, the order of the panels in the legend was incorrect. We have corrected this.

Figure 6H:

Additional information is required. As currently presented in the actual main text body, this tree (or "trees," as mentioned?) does not provide details on the exact sequences used (e.g., GenBank IDs), the species names, or the type of phylogenetic tree. This makes it non-reproducible and limits its usefulness. At the very least, please refer to the supplementary data section where more information might be available.

We apologise for this omission. The information that the referee requests is available in Supplementary Data 12 (Supplementary Data 11 in our previous version), which consists of the full trees with the full names, including accessions, of the sequences used, but we failed to appropriately call this data file in the text. We did it when we referred to the other trees generated, a few words after, but in the revised version we call it twice for clarity.

After all, I would like to emphasize that I consider this manuscript to be of great interest to a broader research community and believe it has the potential to make a valuable contribution to this journal.